# Hyaluronan: An Architect and Integrator for Cancer and Neural Diseases

**DOI:** 10.3390/ijms26115132

**Published:** 2025-05-27

**Authors:** Che-Yu Hsu, Hieu-Huy Nguyen-Tran, Yu-An Chen, Kuan-Ting Lee, Tzong-Yuan Juang, Ming-Fu Chiang, Shin-Yi Liu, Nan-Shan Chang

**Affiliations:** 1Department of Medical Laboratory Science and Biotechnology, College of Medicine, National Cheng Kung University, Tainan 70101, Taiwan; bryanhsuks@gmail.com; 2Graduate Institute of Biomedical Sciences, College of Medicine, China Medical University, Taichung 40402, Taiwan; nguyentranhieuhuy@gmail.com; 3Graduate Institute of Medical Genomics and Proteomics, College of Medicine, National Taiwan University, Taipei 10617, Taiwan; momofish0716@gmail.com; 4Graduate Institute of Medicine, College of Medicine, Kaohsiung Medical University, Kaohsiung 80708, Taiwan; ayta860404@gmail.com; 5Department of Cosmeceutics, China Medical University, Taichung 40402, Taiwan; tyjuang@mail.cmu.edu.tw; 6Department of Neurosurgery, Fu Jen Catholic University Hospital, Taipei 24352, Taiwan; chiang66@gmail.com; 7Department of Biomedical Imaging and Radiological Science, China Medical University, Taichung 40402, Taiwan; 8Department of Neurochemistry, New York State Institute for Basic Research in Developmental Disabilities, New York, NY 10314, USA

**Keywords:** hyaluronan, hyaluronidases, binding proteins, receptors, signaling pathways, cancer progression, neural diseases

## Abstract

Hyaluronan (HA) is essentially secreted by every cell and plays a critical role in maintaining normal cell physiology. While the structure and function of HA have been extensively investigated, questions regarding the sizes and conformation of HA under physiological and inflamed conditions, in relevance to its functions, remain elusive. In this article, we update our knowledge of the HA functional properties, including binding proteins and their signaling networks, as well as matrix formation, which can potentially induce phase separation and affect the mobility and behavior of small molecules, proteins, and cells. We detail the striking differences regarding the biological outcomes of signaling pathways for HA and membrane receptors versus HA and GPI-linked hyaluronidase Hyal-2. We describe: (1) the native, large-sized HA is not proapoptotic but signals with an overexpressed HYAL-2/WWOX/SMAD4 complex to induce apoptosis, which is likely to occur in an inflamed microenvironment; (2) HA-binding proteins are connected via signal pathway networks. The competitive binding of HA and TGF-β to the membrane HYAL-2 and the downstream HYAL-2/WWOX/SMAD4 signaling is addressed; (3) the phase-separated proteins or small molecules in the HA matrices may contribute to the aberrant interactions, leading to inflammation and disease progression; (4) the role of HA and complement C1q in Alzheimer’s disease via connection with a risk factor for Alzheimer’s disease WWOX is also discussed; (5) a hidden function is the inducible HA conformational changes that confer cancer suppression and, probably, retardation of neurodegeneration.

## 1. Hyaluronan: An Architect for Living Cells

### 1.1. In Memory of Gerard Armand, an Outstanding Expert in Hyaluronan

We are currently writing an in-depth review article on hyaluronic acid, also known as hyaluronan (HA). This article serves as a heartfelt tribute to the remarkable discovery of HA—a pivotal achievement in the annals of biochemistry made 91 years ago by Karl Meyer and John Palmer at Columbia University in New York City in 1934. The profound implications of HA for biochemistry continue to captivate and inspire us.

While exploring the captivating story of HA, we were deeply saddened to learn of the recent passing of Professor Gerard Armand from Columbia University. He was not only a cherished long-time collaborator but also a dear friend to Nan-Shan Chang. Professor Armand left us on 16 March 2025, at the age of 93. His lifetime of contributions has greatly enriched our understanding of hyaluronan’s functions, biological properties, and utilization of HA in clinical treatment.

### 1.2. A Brief History of Hyaluronan Use for Eye Surgery and Intraarticular Injection

Hyaluronan does not have a fixed molecular weight. The U.S. Food and Drug Administration (FDA) approved its use in eye surgery and joint injections due to its biocompatibility and biodegradability. Karl Meyer and John Palmer isolated HA from the vitreous body of a cow’s eye in 1934, and, in the same year, Endre Balazs purified HA from umbilical cords and rooster combs. Balazs first conceived of the idea of HA in eye surgery and joint treatment. He conducted the first intravitreous injection of HA in 1958 established the intricate procedure of viscosurgery, subsequently patenting the viscoelastic material forintraocular lenses. Later, Healon was developed by Pharmacia for use in eye surgeries. In the 1970s and 1980s, Healon was widely used in various eye surgeries. In summary, HA benefits eye and knee tissues. Its natural viscoelasticity mimics natural tears and serves as lubrication, and it accelerates wound healing post-surgery in the eye, knee, or other organs.

### 1.3. Brief Introduction to Hyaluronan

We will cover the established knowledge, recent discoveries, and potential future advances for HA. Each cell designs its own microenvironment, a comfortable space for growth and communication with neighboring cells. HA, a key player, can form a robust skeletal framework in the extracellular matrix (ECM). This framework not only supports the cell shape, differentiation, and a crucial function but also filters out unwanted materials while simultaneously facilitating the influx of nutrients and messenger molecules for survival. HA, an anionic, linear, and non-sulfated glycosaminoglycan (GAG), is composed of repeating disaccharide units of D-glucuronic acid and N-acetyl-D-glucosamine. These units are linked through alternating beta-1,3 and beta-1,4 glycosidic bonds [1,2,3,4,5]. The disaccharide unit can repeat 10,000 times or more, building up to sizes of approximately 4 million Daltons. HA’s linear chain can coalesce to form multiple chain cables, establishing a protective network for living cells [1]. For instance, the interchain coupling of HA long chains to form cables by ocular trabecular meshwork cells occurs in the eye under normal conditions and is increased by stress stimuli [2]. This spatial network is protective and ensures a smooth nutrient flow.

The highly hydrophilic nature of HA leads to a significant expansion in volume by 10-fold when fully hydrated. This expansion results in the formation of a mesh-like tertiary structure, primarily due to the formation of β-sheets, which are stabilized by hydrogen bonds. Hyaluronic acid’s ability to form these structures is crucial for its support of various biological functions, including its role as a hydrogel in the body [3]. The hydrogel may incorporate peptides forming β-sheets. This type of hydrogel is suitable for cell culture, tissue engineering, and other uses.

### 1.4. HA Synthesis and Secretion to ECM

HA is present in the dermis, cartilage, vitreous humor, joint synovial fluid, rooster comb, and the cells surrounding oocytes before ovulation in vertebrates [2]. Hyaluronan is involved in fertilization, embryogenesis, development, differentiation, and various cell physiological processes such as cell migration, phagocytosis, and proteoglycan assembly. Its ability to control inflammation, promote wound healing, and influence cancer progression has led to its potential use in drug delivery, cosmetics, and pain relief [3,4]. Mammalian hyaluronan synthases (HAS) synthesize HA into linear long chains, each ranging from 1 million–4 million Daltons. These chains can couple with others to form bundles or cables.

### 1.5. High-Molecular-Weight HA at High Concentrations Is Anti-Inflammatory and Against Cancer Cell Growth In Vivo

Native high-molecular-weight HA (HMW-HA) of 1 million–4 million Daltons forms a porous and flexible matrix to facilitate protein interactions and transportation needed for supporting cell and tissue homeostasis [6]. Involvement of phase separation of biomolecules in the HA matrices is likely. The phase separation allows either enhanced or retarded molecular movement or interactions. Native HA at 1–5 mg/mL levels in synovial fluid, vitreous humor, and other body fluids acts as a barrier, providing anti-inflammatory, anti-angiogenic, and anti-cancer effects. It also plays a crucial role in wound healing. Cancer cells can barely grow in an HA-rich microenvironment. For example, naked mole rats can synthesize a high molecular mass of HA, up to 10 million Daltons. These mice live long lives and have never developed cancer [6]. Conceivably, the higher the HA sizes, the better the suppression of cancer growth. Large-sized HA is less porous at high concentrations, thus limiting nutrient supply to cancer cells. Whether cancer cells undergo apoptosis in the highly concentrated HA is unknown.

For in vitro and in vivo experiments using μg/mL concentrations, native HA is not effective in anti-inflammation, and, worse still, it enhances cancer cell growth [7]. Heparin, a polysulfated glycosaminoglycan at μg/mL levels, has the highest potency in blocking serum-complement activation, surpassing the effects of native HA at the same concentrations. The most effective strategy to enhance the anti-complementary function of HA is to uncouple the chain-associated cables by heating them to 100 °C, thereby exposing the polyanionic charges of the HA long chains. The chain-decoupled HA rapidly binds complement proteins to prevent their activation [8]. However, the heating approach, while straightforward, is not entirely effective, as the heat-induced decoupled chains can rapidly recouple when the temperature drops. Therefore, after heating at 100 °C, quick freezing is necessary to stabilize the chaotic chains. Alternatively, heating HA, followed by freeze-drying, enhances its anti-inflammatory function [8].

### 1.6. HA Sensitizes Cells to Apoptosis by Overexpressed WWOX, HYAL-2, and SMAD4

Without the presence of apoptotic inducers, native HA alone fails to induce apoptosis, although it may exert growth suppression. For example, the high concentrations of HA in the synovial fluids and vitreous humor do not cause apoptosis in the surrounding tissues. Similarly, HA at high concentrations does not induce apoptosis in vitro. To be effective in inducing apoptosis, HA binds membrane-clustered hyaluronidase HYAL-2. Without clustering, HYAL-2 stimulation does not induce cell death. Once native HA is engaged with clustered HYAL-2, the downstream signal goes to tumor suppressors WWOX (WW domain-containing oxidoreductase) and SMAD4. The signaling complex HYAL-2/WWOX/SMAD4 relocates to the nucleus in 20–40 min. If the signal is strong, nucleus-initiated bubble formation and death occur (Figure 1) [5]. If the signal weakens, the HYAL-2/WWOX/SMAD4 signaling supports cell proliferation. When skin keratinocytes are in the basal layer, the signal induces proliferation. When the cells reach cornification, the signal induces cell death by activating WWOX. Notably, when HYAL-2 is replaced by p53 in cells, the cells undergo membrane blebbing without death [5]. We discovered the binding of transforming growth factor beta (TGF-β) to membrane Hyal-2 [9]. Hyaluronan competes with TGF-β in binding to HYAL-2. The biological outcome of the competitive binding, either enhancing cell growth or death, remains to be established.

### 1.7. HA Matrix Provides Phase Separation to Control Molecular and Cell Migration and Mutual Interactions

Every corner of the HA matrix may not be the same. HA phase separation can be achieved by temperature, pH, ionic strength, and chain length. The temperature affects interchain coupling in HA. The strength of interchain coupling is affected by pH, ionic strength, chain density, and length. When a specific drug or molecule is present in the HA matrix, it can freely relocate from one place to another, depending on the nature of each HA matrix. Thus, the occurrence of phase separation in the HA matrix can limit the properties of these drug materials, including self-association, movement, and relocation. The polyanionic HA polymers at high concentrations in vivo may induce phase separation for small molecules, proteins, and DNA. Under this condition, the small molecules are separated in high concentrations (Figure 2). Their movement can be retarded, whereas their interactions remain strong.

HA long interchain cables can be decoupled by the presence of HA short chains or polymers such as the polysulfated chondroitin sulfates or heparin; in this case, it is the so-called microphase separation. The new matrix may or may not facilitate molecular migration. The polysulfation in chondroitin sulfates or heparin will likely alter phase separation and affect the migration of proteins, small molecules, and even cells.

### 1.8. Degraded HA Disrupts the Intact HA Matrix

Full-length HA is subjected to degradation during inflammation and the diseased state in vivo. Low molecular weight HA (LMW-HA; 100–500 kDa) derived from cancer cells promotes cell proliferation, migration, angiogenesis, pro-inflammatory responses, and mesenchymal cell movement [10]. LMW-HA disrupts the intact HA matrix, which affects the normal cell physiology and aberrant cell behaviors [10].

The tumor microenvironment is recognized as a pivotal factor in cancer recurrence and progression. HA accumulation in the stroma and tumor parenchyma is common in cancers like lung, breast, bladder, and prostate, linking to poor clinical outcomes [10]. Several strategies are used to target HA in cancer, including inhibiting HA synthesis and fragmentation, promoting HA degradation, and blocking HA signaling [2]. For the overproduction of HA and HYALs with cancer malignancy and metastatic potential, excessive levels of HASs induce HA overexpression [11]. Furthermore, 4-Methylumbelliferone (4-MU) inhibits HA synthesis by sequestering glucuronic acid and decreasing the expression of HASs [12], thereby decreasing the proliferation, migration, and invasion of cancer cells, as well as reducing tumor growth and metastasis in vivo [13].

Additionally, peritumor HA is digested by HYALs, causing small HA molecules to undergo conformational changes, with long and short chains decoupling and recoupling due to inflammation-induced thermal variations [14]. LMW-HA polymers are likely key stimulators to enhance proliferation, tumor cell migration, invasion, and neo-angiogenesis [10]. It also recruits macrophages, which then polarize into subpopulations that shield tumor cells from adaptive immune attacks [15]. Increased HYAL expression leads to LMW-HA accumulation, observed in colon, prostate cancers, and breast tumor metastasis. In an invasive bladder cancer cell line, blocking HYAL expression suppresses tumor growth, inhibits tumor infiltration, and decreases microvessel density [16]. Conversely, reduced HYAL expression may lead to higher mortality rates in ovarian, pancreatic, and endometrial cancers. The finding suggests that HYALs’ maladaptive effects in cancer may vary by tissue types [17]. Therefore, HYALs could be targeted in oncologic therapies, tailored to the specific pathogenesis of each cancer [18].

## 2. HA Biosynthesis and Catabolism

### 2.1. HA Synthesis by Hyaluronan Synthases (HASs)

HA is synthesized on the inner side of the plasma membrane by Class I hyaluronan synthases HASs, which are lipid-dependent integral membrane glycosyltransferases. These enzymes have eight membrane domains (MDs) in vertebrates and six MDs in Streptococcal species, enabling the initiation, elongation, and translocation of HA into the ECM [18,19]. Class I HASs are structurally similar, with high sequence identity, comparable membrane-domain organizations (6–8 MDs), topologies, and processive mechanisms [19]. In mammals, three individual HAS genes are responsible for encoding the HA-synthesizing isoenzymes: HAS1, HAS2, and HAS3 [19,20]. These isoenzymes are expressed at specific times and in specific tissues during development, aging, wound healing, and under normal and pathological conditions, including diseases such as cancer. The only known Class II HAS, from *Pasteurella multocida* (PmHAS), differs from Class I HASs in membrane attachment, sequence, domain organization, and its distributive (non-progressive) mechanism [21]. Moreover, PmHAS catalyzes the elongation of HA by sequentially adding single monosaccharides to the non-reducing end of the HA chain [22].

HAS genes and their proteins exhibit distinct patterns of expression and activity across different times and locations. The first identified mammalian HAS gene, HAS1, produces a full-length transcript (HAS1-FL) and three variants (HAS1-Va, HAS1-Vb, HAS1-Vc), which are linked to poor survival in malignant myeloma and Waldenström’s macroglobulinemia. The HAS2 gene comprises one protein-coding transcript and a regulatory antisense RNA (HAS2-AS1), while the HAS3 gene comprises five protein-coding transcripts (HAS3-v1, HAS3-vX1, HAS3-v2, HAS3-vX2, HAS3-v3). HAS2 and HAS3 expressions are associated with malignant transformation. HAS1 is primarily localized in the cytoplasm, with only a small fraction of ectopically overexpressed HAS1 found at the plasma membrane [19,20,21], whereas HAS2 and HAS3 are found in both the plasma membrane and cytoplasm.

HASs do not require an exogenous primer to initiate HA synthesis. They polymerize the GlcNAc(β1,4)GlcUA(β1,3) disaccharide by adding the two uridine diphosphate (UDP)-sugar substrates, UDP-α-N-acetyl-D-glucosamine (UDP-GlcNAc), and UDP-α-D-glucuronic acid (UDP-GlcUA) at the reducing end through two glycosyl-transferase activities. These activities create two different β-1,4 and β-1,3 glucuronide bonds between sugar monomers to elongate the linear polysaccharide HA [19,22,23,24]. UDP-GlcNAc and UDP-GlcUA are derived from glycolytic intermediates, glucose-6-phosphate, and fructose-6-phosphate, respectively, suggesting a potential connection between high glucose uptake and excessive HA accumulation in various tumors. During HA synthesis, Class I HASs perform three fundamental functions: (1) synthesizing disaccharide units to form HA, (2) creating and releasing HA chains of specific sizes or lengths, and (3) translocating the growing chain across the cell membrane to the exterior simultaneously with elongation [22,23,24].

The effect of HAS1 mainly depends on its dependence on UDP–sugar availability and its role in regulating cellular metabolism and morphology [16]. HAS1 requires higher cellular UDP–sugar concentrations compared to HAS2 and HAS3 for hyaluronan synthesis. This discovery highlights the crucial interplay between HAS1 activity, cellular UDP–sugar levels, and HA synthesis.

HAS2 produces the largest HA polymers, exceeding 2 × 10^6^ Da, while HAS1 produces polymers in the range of 2 × 10^5^–2 × 10^6^ Da, and HAS3 produces polymers ranging from 1 × 10^5^–1 × 10^6^ Da [18]. HAS1 is the least catalytically active among the HASs, while HAS2 is inherently less active than HAS3. Kinetic analyses also reveal that HAS1 has higher Km values for UDP-GlcUA and UDP-GlcNAc than both HAS2 and HAS3 [19]. The activities of HASs are modulated by post-translational modifications, such as phosphorylation, ubiquitination, and O-GlcNAcylation. These activities are lipid-dependent and regulated by their lipid and cholesterol microenvironment [20,22]. Abnormal HAS activation may occur, thereby increasing HA production during embryogenesis, inflammation, and cancer growth. HAS1 is frequently elevated in inflammatory diseases like atherosclerosis, osteoarthritis, and lung infections, offering potential for better understanding and treatment of these conditions [23]. Has2 is crucial for embryonic development, as shown in Has knockout mice, while Has1 and Has3 have no impact. The heterodimeric complexes involving HAS1-HAS2, HAS2-HAS2, and HAS2-HAS3 have been reported [25].

A most advanced feature is that HASs can add disaccharide units to the growing hyaluronan chain either at the reducing end or the non-reducing end [20,21]. This discovery expanded the understanding of how HASs function, moving beyond the traditional view of sugar addition at the non-reducing end. The presence of intracellular hyaluronan suggests that active HASs might be found within cells, not just on the cell membrane. 

### 2.2. Hyaluronidases in HA Catabolism

The catabolism of HA is a complex process, involving HYALs, reactive oxygen/nitrative species (ROS/NOS), access to lymph, and cell-local uptake [26]. Interestingly, human HYALs are found in various organs (testis, spleen, skin, eyes, liver, kidneys, uterus, and placenta) and body fluids (tears, blood, and semen), highlighting the widespread presence of these enzymes. They cleave the β-1,4-glucosaminidic bond between glucosamine and glucuronic acid [27]. Six hyaluronidase-like genes have been identified in humans: HYAL1, HYAL2, HYAL3, HYAL4, PH-20/SPAM1 (sperm adhesion molecule 1), and a pseudogene HYALP1 [28]. HYAL1, HYAL2, and HYAL3 cluster on the chromosome 3p21.3, and HYAL4, HYALP1, and SPAM1 are located on chromosome 7p31.3.

HYAL1 and HYAL2 have been shown to play significant roles in the tumor microenvironment [29]. HYAL1 is a lysosomal enzyme that can be released from cells and found in tissues, circulation, and secretion [30]. It is also internalized by monocytes and endothelial cells relocating to lysosomes, and it remains functionally active at low pH (pH = 3.8) [28,31]. Furthermore, HYAL1 has been observed to play an important role in regulating ovarian folliculogenesis by modulating the follistatin/activin/Smad3 pathway [31]. HYAL2 is not only a lysosomal protein whose enzymatic activity can be induced under acidic environments, but it is also a potential tumor suppressor [32]. A 20-amino-acid signal sequence at the C-terminus allows for the formation of a glycosylphosphatidylinositol (GPI) linkage, anchoring it to the cell membrane [33]. Functionally, membrane HYAL2 degrades HA bound to CD44, generating approximately 20 kDa fragments. Essentially, surface HYAL2 acts as a co-receptor with CD44 for HA, and CD44 aids HYAL2 in the degradation of HA [34]. HA fragments can undergo endocytosis, after which HYAL1 continues to degrade these fragments [30]. In addition, HYAL2 serves as a receptor for jaagsiekte sheep retrovirus, contributing to glycalyx formation [33,35]. No HYAL2 deficiency has been identified in humans. In Hyal2-deficient mice, exceptionally large HA molecules accumulate in tissues and circulation, leading to chronic thrombotic microangiopathy and mild craniofacial and vertebral abnormalities [36,37]. Megakaryocyte-derived HYAL2 facilitates HA degradation, which is essential for thrombopoiesis [38].

HYAL3, despite being enzymatically inactive, plays a vital role in supporting the activity of HYAL1 [39]. Hyal3 knockout mice do not show evidence of hyaluronan accumulation [40]. HYAL3 is predominantly found in the testis, with enzymatic activity detected in spermatozoa [41]. HYAL4, although less understood, has limited expression and may function as a chondroitinase rather than a hyaluronidase [28]. PH-20, also known as SPAM1, is primarily found in the testis and sperm, and it exhibits activity at higher pH levels. As a membrane GPI-anchored protein, PH-20 is released in a soluble form and is expressed in the lysosome-derived acrosome on a sperm’s surface. PH-20 aids fertilization by enhancing sperm penetration to the ovum through extracellular matrix cleavage [26]. PH-20 is also expressed in breast and other cancer tissues and is recognized as a tumor marker for laryngeal cancers [42]. The application of recombinant human PH-20 in clinical settings has shown potential in enhancing drug penetration to cancer targets, offering a hopeful avenue for cancer treatment [43].

A deficiency in HAS and HYALs contributes to various types of diseases. For example, mucopolysaccharidosis IX results from the lack of HYAL1 and HAS2 and is associated with cardiac pathology in affected individuals [44]. These enzymes have been found to have extensive use in clinical applications. HYALs are used as adjuvants to break down the extracellular matrix, improving drug penetration to target areas [45].

## 3. HA-Interacting Partners

### 3.1. Hyaluronan-Binding Proteins and Receptors

This section provides rationales linking HA with its cognate receptors or binding proteins (HABP; also known as hyaladherins). Some of these proteins or receptors drive and maintain the biological functions or pathological alterations in each cell. These proteins can be intracellular or extracellular. HA is synthesized essentially by every cell. HA is integrated into structural components of the connective tissues and matrices to support or limit cell division and migration, immune cell adhesion and activation, and intracellular signaling [46,47]. The complicated roles of HA in vivo stem from its large number of hyaluronan-binding proteins that render significant differences in signaling and differential gene expression in tissues.

Many HABPs share a Link module of 100 amino acids for ligand binding. The Link module for HABPs is considered a superfamily. The Link protein consists of an immunoglobulin domain, two Link modules, and a structure found in the G1 domains of aggrecan, versican, neurocan, and brevican. This configuration, first identified in cartilage, includes two α-helices and two triple-stranded β-sheets. These proteoglycans form large HA-linked complexes, contributing to cartilage’s load-bearing, blood vessels’ elasticity, as well as the structural strength of tissues like skin and brain [48]. Subsequently, we will discuss HABPs, including HA receptors and the extracellular molecules associated with HA.

### 3.2. HA Receptors and Functions

HA is also localized in several intracellular organelles, such as the endoplasmic reticulum, nucleus, centrosomes, and other cytoplasmic compartments. The origin and functions of intracellular HA remain unclear. Receptor-mediated endocytosis is likely the main route for HA’s intracellular localization, differing from its rapid turnover. Binding proteins for HA that have been studied extensively are: cluster of differentiation 44 (CD44), receptor for hyaluronan-mediated motility (RHAMM), lymphatic vessel endothelial receptor for hyaluronan 1 (LYVE1), hyaluronan receptor for endocytosis (HARE), and innate immune receptors such as toll-like receptors 2, 4, and 5 (TLR2, TLR4, TLR5) [3,49,50]. Less studied receptors for HA include layilin (LAYN), hyaluronan-binding protein 1 (HABP1/C1QBP/gC1qR/p32), intracellular adhesion molecule-1 (ICAM-1), etc [51]. These receptor proteins generally participate in inflammation and cancer motility. They also play a vital role in developing stem cells and cancer stem cells by regulating cell proliferation and differentiation. Networking analysis reveals the protein/protein-interacting network for HA-binding proteins. Two representative layers of networking are shown (Figure 3). Participation of additional proteins, reaching thousands or more components, in the networking is anticipated.

Based on the networking, outstanding questions that remain to be solved are (1) whether HA long cables or cable-like materials with similar conformation bind all the receptors or binding proteins, (2) do microenvironmental changes alter the conformation and the binding affinity of HA with each receptor? (3) how do the signaling pathways operate among all the receptor proteins—in a nicely integrated manner or in conflict?

### 3.3. Cluster of Differentiation 44 (CD44)

CD44, a single-span transmembrane glycoprotein, is the most widely recognized HA receptor. CD44 is essential in maintaining the normal homeostasis of various tissues. CD44 is highly expressed in cancer cells and serves as a key marker for cancer stem cells (CSCs) [52]. The human CD44 gene, located on chromosome 11p13, contains 19 exons, while the mouse CD44 gene has 20 exons, with an additional exon V1 not present in humans. Exons 1–5 encode the extracellular region, exons 16–17 encode the stalk region, exon 18 encodes the transmembrane domain, and exons 19–20 encode the intracellular domain. The standard CD44 isoform (CD44s) consists of exons 1–5 and 16–20 and represents the smallest isoform with 365 amino acids. Alternative splicing of exons 6–15 generates variable regions that are inserted between the extracellular common region and stalk region, leading to the formation of multiple CD44 variant isoforms (CD44v). Different variable regions provide new conformations or binding sites to CD44v, enabling its diverse functions [53]. All CD44 isoforms consist of one distal extracellular domain that binds to extracellular components, including its classical ligands HA and osteopontin (OPN, phosphoproteins). They also feature a transmembrane domain and a cytoplasmic domain, which contains binding motifs for cytoskeleton-related proteins (e.g., ankyrin and cytoplasmic tyrosine kinases) [54]. Moreover, post-translational modifications of CD44, such as glycosylation, frequently become deregulated in cancer. For instance, truncated O-glycosylation enhances HA’s binding to CD44, promoting tumorigenic signaling.

HA, a key component of the ECM, is internalized and distributed within distinct cytoplasmic vesicles in cells. HA serves as a CD44 ligand, with its endocytosis being size-dependent, as different HA lengths impact CD44 clustering [9]. Fragmented HA is internalized, whereas HMW-HA is not [55]. When HMW-HA binds to CD44, it is first broken down by GPI-linked HYAL2 through interaction with the Na^+^-H^+^ exchanger 1 (NHE1), creating an acidic environment that activates HYAL2 [56]. The intracellular domain of CD44 seems to be essential for the endocytosis of HA. Due to its ability to bind ankyrin, the intracellular domain of CD44 facilitates CD44-HA endocytosis through a spectrin-mediated uptake of ankyrin-binding proteins [57]. The intracellular domain of CD44 interacts with EH-containing domain 2 (EHD2), a lipid raft protein that connects endocytosis to the actin cytoskeleton, along with other proteins like ERM molecules, contributing to the regulation of HA uptake [58].

After HA endocytosis, Hook1 controls CD44 recycling to the cell surface by aiding the microtubule-dependent sorting of CD44 from EAA1^+^ endosomes into recycling tubules [59]. EHD proteins like EHD2 also control this recycling process through interaction with the CD44 intracellular domain [58]. Ubiquitination of CD44’s intracellular domain presents another potential mechanism for CD44-mediated HA endocytosis [60]. In this situation, CD44 is recycled to the cell surface through clathrin-independent carriers after HA endocytosis. CD44 undergoes ubiquitination by the membrane-associated RING-CH (MARCH)-VIII ubiquitin ligase, causing it to move to EAA1^+^ compartments and late endosomes/lysosomes for degradation [61]. Following endocytosis, HA is gradually broken down by enzymes like HYAL1, β-endoglycosidase, β-exoglycosidases, β-glucuronidase, and β-N-acetylglucosaminidase into tetrasaccharides and, eventually, into individual sugars in lysosomes. These sugars are then transported out of lysosomes into the cytoplasm for energy or biosynthesis. CD44 also regulates cytosolic UDP-GlcNAc levels and plays a key role in HA metabolism, particularly in relation to glucose consumption [62]. High glucose levels induce pericellular HA synthesis by enhancing cytosolic UDP-GlcNAc through the hexosamine biosynthetic pathway, a key nutrient sensor. HA is subsequently internalized through a CD44-mediated pathway. Therefore, excessive glucose may cause elevated cytosolic UDP-GlcNAc, leading to abnormal O-GlcNAcylation of cytoplasmic proteins, including those involved in diabetic complications. Consequently, CD44-mediated HA endocytosis and metabolism may be essential for maintaining homeostasis.

### 3.4. Receptor for Hyaluronan-Mediated Motility (RHAMM)

RHAMM, or the cluster of differentiation 168 (CD168), also called intracellular hyaluronan receptor (IHABP) and hyaluronan-mediated motility receptor (HMMR), is rarely found in normal tissues [63]. However, its expression is significantly induced upon inflammatory stimuli [64,65]. One of its primary functions is in tissue repair, where it is temporarily upregulated in fibroblasts following injury, thereby facilitating the healing process. It also aids neural development by promoting the division of neuroprogenitor cells [46]. RHAMM binds to both fragmented HA and HMW-HA and collaborates with CD44 in HA endocytosis and signaling [66]. In non-adherent cells, RHAMM primarily facilitates HA uptake, whereas in adherent cells, CD44 serves as the principal HA endocytic receptor [67]. Additionally, RHAMM is often overexpressed in human tumors, which correlates with metastasis, aggressive traits, and worse prognosis, making it an independent predictor of cancer outcomes.

Human RHAMM, a helical glycoprotein encoded by the *HMMR* gene on chromosome 5 (5q33.2-qter), consists of 18 exons and 2 start codons [68,69]. Its full-length form weighs 84 kDa and has 725 amino acids, with isoforms arising from alternative splicing, start codons, and post-translational modifications [70]. RHAMM is a hydrophilic protein with three key domains: an amino-terminal domain (approximately 163 amino acids), a rod-like middle supercoiled-coil domain (comprising 5 coils), and a carboxyl-terminal domain (around 11 amino acids) [68,69,70]. The supercoiled-coil domain mediates protein–protein interactions [70]. RHAMM affects cell migration through binding to receptors like CD44 and platelet-derived growth factor receptor (PDGFR), as well as interacting with intracellular proteins such as breast cancer gene 1 (BRCA1), targeting protein for Xklp2 (TPX2) and extracellular signal-regulated kinase 1/2 (ERK1/2), which are involved in mitotic spindle formation and stability. RHAMM possesses an HA-binding domain at the C-terminus, specifically located on amino acids 635–646 and 657–666 [46]. Although RHAMM is an HA-binding protein, it lacks the typical HA-binding motif (proteoglycan tandem repeat). Instead, it interacts with HA through a basic amino acid-rich domain, B(X7)B, where B represents lysine or arginine, and X is a non-acidic amino acid [68,69]. Additionally, RHAMM lacks a membrane-spanning domain, which allows it to be soluble and localized in various cellular compartments, including the cytoplasm, nucleus, cell membrane, and/or the extracellular matrix (ECM) [70]. Therefore, RHAMM is a multifunctional protein that plays a role in various signaling processes both at the cell surface and within the cell.

Cell surface RHAMM regulates HA-induced cell migration, which is crucial for inflammation and wound healing [65,68]. RHAMM/HA complexes on the cell surface often incorporate other proteins, such as protein kinase receptors (RTKs) and non-RTKs, including CD44, PDGF, transforming growth factor beta (TGF*β*), or Recepteur d’origine nantais (RON) [71,72]. The different compositions of these complexes affect downstream signaling by triggering various molecular switchers such as Src and Ras [71]. Intracellular RHAMM (iRHAMM), a protein involved in the cell cycle, contributes to the formation of mitotic spindles and microtubules, and it is present in both the cytoskeleton and nucleus. Due to its interactions with various kinases, iRHAMM is believed to connect the cytoskeleton with signaling complexes, impacting cell motility and proliferation [70]. Numerous studies have demonstrated the significant role of RHAMM in the development and progression of several cancers [65]. RHAMM’s oncogenic potential is linked to its involvement in cell cycle progression, particularly in mitotic spindle formation and stability, as well as its extracellular role in cell migration. It is constitutively expressed in several carcinomas, such as breast, lung, gastrointestinal tract, and prostate cancers, as well as in the aggressive forms of lymphomas, leukemias, and multiple myeloma. RHAMM variants play diverse roles in cancer. For instance, the overexpression of an RHAMM isoform that lacks the N-terminal microtubule-binding domain enhances pancreatic cancer development in both mouse and xenograft models [71]. However, recent studies reveal that RHAMM has varying roles in different cancer subtypes. In aggressive breast cancer, RHAMM overexpression is associated with higher motility and invasiveness, while in the luminal A subtype, it suppresses cell migration [73].

### 3.5. Lymphatic Vessel Endothelial Hyaluronan Receptor-1 (LYVE-1)

LYVE-1, also known as an extracellular link domain containing 1 (XLKD1) and cell surface retention sequence binding protein-1 (CRSBP1), is encoded by the *LYVE1* gene in humans and is a CD44 homolog sharing approximately 44% similarity to CD44. LYVE-1 is a single-span transmembrane protein with an extracellular domain featuring a highly conserved link motif, a transmembrane domain, and a brief cytoplasmic tail similar to CD44. Its stalk region connecting the transmembrane domain and the link module is extensive O-glycosylation [74]. LYVE-1 is primarily expressed by lymph vessels as an HA receptor and serves as a specific marker to distinguish between lymphatic and blood vessels [74]. Mechanistically, LYVE-1 functions as a surface receptor, a signal transducer, or a decoy for HA [49,75,76]. LYVE-1 is involved in the uptake and degradation of HA within lymphatic endothelial cells, most likely for catabolism [74]. It also facilitates the transport of HA into the interior of afferent lymphatic vessels, where it is reabsorbed and broken down in lymph nodes. The precise mechanism of LYVE-1 endocytosis is still unclear, though its association with lipid rafts appears to be crucial for this process [77]. Notably, lymph nodes are crucial for HA removal, as only around 5% is cleared by the hyaluronan receptor for endocytosis (HARE) in the liver.

LYVE-1 possesses low affinity binding to HA, and its efficient binding to HA chains is greatly enhanced through dimerization or removal of extensive sialylation [49,74]. The LYVE-1 binding domain, functioning as a monomer, has a weaker affinity for HA (Kd of 35.6 μM) than the CD44 domain (Kd of 65.7 μM) [78]. LYVE-1 dimers, crucial for binding, typically interact with HA of intermediate-to-high molecular weight and facilitate its uptake. This protein binds HA through a conserved domain within the HABP superfamily. Unlike HA/CD44 interactions, which rely on hydrogen bonds and hydrophobic forces, HA/LYVE-1 interactions are predominantly electrostatic, making them sensitive to ionic strength. LYVE-1 can bind both soluble LMW HA and HA embedded in the pericellular matrix [49]. The binding efficiency is higher when LYVE-1 is densely clustered or in the presence of HMW-HA, which enhances binding strength [79].

LYVE-1 is frequently overexpressed in various cancers, where it is considered a poor prognostic indicator [80]. While the exact role of LYVE-1 in cancer progression is not fully understood, some evidence points to its involvement in lymphangiogenesis and the development of lymph node metastases. The binding of LMW-HA to LYVE-1 on lymph node endothelial cells promotes cell proliferation, migration, and tube formation by activating tyrosine phosphorylation of protein kinase C*α*/*β*II (PKC*α*/*β*II) and ERK-1,2 [75]. The LMW-HA/LYVE-1 complexes also interact with vascular endothelial growth factor C (VEGF-C) and fibroblast growth factor 2 (FGF-2) to promote lymphatic endothelial cell growth and lymphangiogenesis [81]. Blocking HA/LYVE-1 interactions with antibodies suppresses cell growth and movement, lowers lymphatic vessel numbers, decreases primary tumor size, and prevents lymph node metastasis in breast cancer models in mice [75]. The mechanism by which LMW-HA enhances lymph node metastasis formation likely involves changes in lymphatic vessel permeability [80]. Interaction of LMW-HA with LYVE-1 in human dermal lymphatic endothelial cells alters VE-cadherin and β-catenin at membrane junctions, affecting the lymphatic lumen’s integrity.

Unlike the full receptor, the cleaved ectodomain of LYVE-1 acts as a decoy receptor for LMW-HA fragments, blocking cell growth, migration, and lymphangiogenesis [76]. Soluble LYVE-1 (sLYVE-1) is generated locally through MT1-MMP activity and then released into the tumor stroma [82]. sLYVE-1 blocks PI3Kδ signaling, NF-κB activation, and VEGF-C production, thereby reducing lymphangiogenesis. However, sLYVE-1 produced by M2-like tumor-associated macrophages during early cancer stages helps resolve inflammation by clearing LMW-HA from the tumor microenvironment. The sLYVE-1/LMW-HA complexes prevent cell growth in early-stage human and mouse melanomas but have no effect in later stages [76].

### 3.6. Hyaluronan Receptor for Endocytosis (HARE)

HARE, or Stabilin-2 (Stab-2), encoded by the *STAB2* gene in humans, is a type I single-pass membrane protein that exists in two isoforms [66]. The full-length 315 kDa protein can be cleaved to produce a 190 kDa variant [83]. Both isoforms contribute to HA uptake, and varying isoform ratios have been noted in the spleen and liver, but the regulatory mechanisms behind their expression are not yet understood [83].

HARE contains four endocytic-signaling motifs that are mediated by AP-2/clathrin [84]. Most of HARE is localized intracellularly, with only 30–50% found on the cell surface, indicating its temporary presence on the membrane, typical of an endocytic receptor [83]. HARE is primarily located in the endocytic and recycling compartments of cells within lymph vessels and nodes, liver, and spleen [79]. HARE binds to HA via its link domain and facilitates HA uptake through clathrin-mediated endocytosis, promoting the quick removal of HA from biological fluids [85]. The binding and endocytosis of 40–400 kDa HA by HARE can trigger MAPK, ERK-1/2, and NF-κB pathways, leading to gene expression [50]. The precise downstream-signaling pathways and their effects on cell behavior are not yet fully understood, but sensitivity to HA size may be connected to extracellular matrix turnover. Additionally, HARE functions as a scavenger receptor for glycosaminoglycans, low-density lipoprotein particles, phosphatidylserine, and other biological molecules resulting from matrix breakdown [86].

In cancer, inhibiting or removing this receptor blocks HA absorption and the development of metastases [85,87]. Two potential mechanisms have been proposed for this response: (1) the accumulation of HA fragments prevents circulating cancer cells from interacting with nearby tissues [87], or (2) HARE could function as an endothelial receptor for metastatic tumor cells coated with HA, aiding their invasion into tissues [85]. Cell coculture studies have confirmed HARE’s involvement in the interaction between invasive breast cancer cells and lymphatic endothelial cells [88]. Altered gene expression in lymphatic endothelial cells was observed only when co-cultured with aggressive metastatic breast and prostate cancer cells. The alterations in gene expression involved increased levels of metastasis-related genes, including those regulating the cell cycle (CDC6, CLSPN, kinases genes), cell adhesion and movement (BST2, SELE, and HMMR), cytokines (CCL7, CXCL6, CXCL1, and CSF1), and complement system components (C1R, C3, and CFB), all of which play a role in lymphangiogenesis. Moreover, HARE downregulation in cocultures potentially prevented HA absorption from the tumor microenvironment via clathrin-mediated endocytosis, leading to elevated HA levels in the tumor.

### 3.7. Toll-like Receptors (TLRs)

TLRs are type I transmembrane receptors, consisting of 700–1100 amino acids, found on immune cells, including macrophages and dendritic cells. They are crucial for both innate immunity and the activation of adaptive immune responses. TLRs contain extracellular leucine-rich regions and a cytoplasmic domain responsible for recognizing pathogen-associated (PAMPs) and damage-associated molecular patterns (DAMPs), like HA fragments, initiating inflammation [89].

TLRs function as dimers, either homodimers or heterodimers with other TLRs or proteins, thus increasing ligand diversity. In humans, TLRs are grouped into cell-surface TLRs (TLR1, TLR2, TLR4, TLR5, TLR6) for detecting bacterial components, and endosomal TLRs (TLR3, TLR7, TLR8, TLR9) for recognizing microbial or viral nucleic acids [90,91]. TLR2 and TLR4 bind HA, and the binding depends on HA molecular size. At least four monosaccharide units (4-mer) of HA are needed for TLR2 or TLR4 binding, while longer fragments (6-mer) enable TLR4-CD44 heterodimer formation, thereby enhancing intracellular signaling [92].

TLRs are expressed in various cancers, such as hepatocellular carcinoma, melanoma, neuroblastoma, and cancer cells originating from the lung, colon, breast, ovary, cervix, and prostate [93]. In later stages of cancer, HA/TLR2,4 signaling promotes tumor cell invasion and cytokine/chemokine expression, contributing to cancer progression. The impact of HA/TLR2,4 varies with the cell type and tumor environment.

### 3.8. HA Association with Extracellular Molecules

HA is not found as an isolated molecule in the ECM. For decades, it has been recognized that ECM HA is covalently bound to other proteins or glycoproteins, collectively known as “hyaladherins or HABPs”. Notably, HA-signaling properties are present only when bound to HABPs, not in their free form. This review will focus on four key HABPs: inter-α-inhibitor (IαI), tumor necrosis factor-stimulated gene 6 (TSG-6), pentraxin-3 (PTX3), and versican, because their interactions with HA are well characterized and offer clear therapeutic potential.

#### 3.8.1. Inter-α-Trypsin Inhibitor (IαI)

IαIs are a group of naturally occurring and structurally similar molecules found abundantly in human plasma and various cell types. It comprises a common light chain (LC) and multiple similar heavy chains (HCs). The LC features a distinctive combination of a chondroitin 4-sulfate domain and a serine protease inhibitor core, known as bikunin, ulinastatin, or urinary trypsin inhibitor (UTI). Chondroitin sulfate connects the LCs and HCs through a distinctive protein–glycosaminoglycan–protein crosslink. HCs mainly serve to stabilize the ECM. IαIs in plasma mainly exist as an inter-alpha inhibitor (IαI), made of two HCs (HC1, HC2) and one LC, and a pre-alpha inhibitor (PαI), containing one HC (HC3) and one LC. These IαI proteins are mainly synthesized in the liver, where they are paired with bikunin before being released into the bloodstream. However, they can also be produced in other solid organs, such as the lungs, kidneys, and brain [94,95]. Collectively, IαIs are immunomodulatory molecules that reduce protease damage, suppress pro-inflammatory cytokines, enhance anti-inflammatory cytokines, modify immune cell function, decrease complement activation, and bind histones in systemic inflammation. The binding of IαI-HCs to HA through a transesterification reaction is a critical process. This reaction is catalyzed by a 35 kDa-secreted ECM HABP, tumor necrosis factor-stimulated gene 6 (TSG-6). Triggered by TNFα and released by immune and structural cells, TSG-6 is crucial in inflammation. Its induction, along with IαI production and extravasation, promotes the formation of a “pathological” HA-HC matrix [27,96].

#### 3.8.2. Tumor Necrosis Factor-Stimulated Gene 6 (TSG-6)

TSG-6 and IαI exhibit intriguing paradoxical effects in the development of inflammation. Both proteins are elevated in conditions like asthma, COPD, cystic fibrosis, lung fibrosis, atherosclerosis, and kidney disease [27,95,96]. Firstly, these proteins are essential for generating a pathological HA matrix and developing airway inflammation and hyperresponsiveness in allergic and TLR4-mediated lung injury [96]. Conversely, they possess significant anti-inflammatory properties in other disease models, including lung inflammation following endotoxin and bleomycin exposure, pancreatitis, arthritis, and various other conditions [95,97]. TSG-6 is believed to promote the beneficial effects of stem cells in regenerative models. IαI binds to coagulation factor IX and acts as a potent inhibitor of factor XI [98,99]. IαI inhibits plasmin, significantly improving disseminated intravascular coagulation and reducing lung injury in LPS-induced sepsis [100,101]. Additionally, IαI inhibits complement activation, thereby reducing complement-related lung injury [102]. Ultimately, IαI can directly bind to pathogen-associated molecular patterns like the dengue virus, adenovirus, and HIV (84, 85), as well as danger-associated molecular patterns such as histones and HMGB1 [103,104]. IαI suppresses inflammation by binding infectious and danger-associated molecular patterns.

#### 3.8.3. Pentraxin-3 (PTX3)

In the HA matrix, PTX3 crosslinks IαI HCs [105]. PTX3, IαI, and TSG-6 are essential for forming a functional HA matrix in the ovulating ovary’s cumulus oophorus, with their absence causing infertility. PTX3, induced by TLR activation during inflammation, aids in injury repair, antimicrobial defense, and allograft survival [106,107,108]. PTX3 also mitigates allergic lung inflammation and LPS-induced lung injury (like TSG-6 and IαI) [109,110]. Additionally, it aids in injury repair, fibrin deposition, and fibrosis in an acid aspiration lung injury model. Like IαI, PTX3 binds viral antigens such as influenza hemagglutinin and interacts with the complement system, though its impact on activation remains unclear [111].

The complex and conflicting roles of TSG-6, IαI, and PTX3 in inflammation are better understood by examining the specific inflammatory mechanisms of each disease or model. Pathological HA matrix formation is essential for disease development, such as airway hyperresponsiveness following lung injury or allograft dysfunction [96]. PTX3, IαI, and TSG-6 contribute to the disease by facilitating the creation of the pathological HA matrix. Conversely, in processes not mediated by HA, HA matrices supported by PTX3, TSG-6, and IαI can have an anti-inflammatory effect by mitigating pathways dependent on complement or coagulation factors [97]. Typically, acute conditions benefit from LMW-HA matrix accumulation, which enhances inflammation and helps clear harmful agents. In contrast, chronic or degenerative conditions, lacking homeostasis, may suffer from excessive HA matrix accumulation.

#### 3.8.4. Versican

HA is further modified by the proteoglycan versican, which helps regulate the inflammatory response through a feedback loop [112,113]. Versican is produced in response to TLR activation, infection, or aseptic injury, and it controls the activity of chemokines, immune cell retention, and chemotaxis [68,112,114]. Like HAS2, versican has differing effects depending on the cell type: it can be anti-inflammatory when expressed in macrophages or pro-inflammatory when expressed in mesenchymal cells [68,114]. Collectively, research indicates that the HA matrix, comprising cross-linked TSG-6, PTX3, IαI HC, and versican, regulates immune responses to injury in complex and cell-specific manners. In fact, understanding HA biology necessitates analyzing HA’s interactions with HABPs. The HA matrix acts as a dynamic canvas decorated by HABPs, and these interactions likely dictate HA’s specific effects and roles following injury [113].

## 4. The Role of HA-HABP Interactions in Human Diseases

Overall, HA binds to cell surface receptors like CD44, RHAMM, LYVE1, and HARE, as well as innate immune receptors TLR2, TLR4, and TLR5 [115]. Its size is crucial for receptor signaling, as longer HA chains provide multiple binding sites, promoting receptor clustering [116]. Typically, HMW-HA is anti-inflammatory and anti-angiogenic, whereas LMW-HA is pro-inflammatory and pro-angiogenic, contributing to tumor progression [116,117,118]. Interestingly, hyaluronan oligosaccharides (oHA) can either promote or suppress inflammation, depending on the cell type and disease model. While HA signaling usually begins at the cell membrane, intracellular HA also exists. Intracellular HA may be involved in physiological processes, such as interacting with the mitotic spindle, microtubules, and RHAMM receptor during mitosis. In pathological conditions, such as hyperglycemia, HA may be abnormally produced in the Golgi, leading to endoplasmic reticulum stress and triggering an autophagy response.

HA metabolism dysregulation is a hallmark in many inflammatory and degenerative diseases. Elevated HA levels in circulation or tissue are observed in lung disorders like asthma, COPD, interstitial lung disease, and pulmonary hypertension, as well as in inflammatory, fibrotic, and degenerative diseases such as rheumatoid arthritis, hepatitis, cirrhosis, chronic kidney disease, atherosclerosis, cardiac remodeling in heart failure, diabetic pancreas, allograft rejection, and most solid cancers [12,116,119,120,121,122]. In many of these diseases, higher serum HA levels are associated with worse outcomes and prognosis, with LMW-HA being the most common form detected. Furthermore, abnormal HAS expression is frequently observed in cancers and tissue injuries, correlating with clinical outcomes. Collectively, the dominant theory suggests that elevated HA expression in the tumor stroma drives cancer progression and metastasis by promoting pro-proliferation signals, supporting immune and apoptosis evasion, facilitating angiogenesis and lymphangiogenesis, increasing invasiveness and metastatic potential, and reprogramming energy metabolism [123].

Abnormal HA matrix deposition, often linked with HABPs, occurs in several pathological processes involving tissue remodeling, such as pulmonary hypertension, inflammatory bowel disease, and asthma [27,124]. Under endoplasmic reticulum stress, cells generate “HA cables” (HA with IαI HC, and sometimes TSG-6) that extend from the perinuclear region into the extracellular space, creating an adhesive matrix that traps inflammatory cells [125]. Versican deposition is elevated in the airways of patients with idiopathic pulmonary fibrosis, severe ARDS, and asthma. In lung transplant patients, higher PTX3 levels are linked to graft dysfunction, including ischemia-reperfusion injury [126,127], while genetic variations in PTX3 that increase serum PTX3 levels predispose individuals to allograft dysfunction [108]. Conversely, genetic variations in PTX3 that reduce its function are associated with a higher risk of invasive aspergillosis in stem cell transplant recipients [107]. In osteoarthritis, TSG-6 levels in synovial fluid correlate with disease progression [128]. Additionally, circulating IαI levels are associated with survival in infections and inflammatory diseases such as severe sepsis in adults, necrotizing enterocolitis in newborns, and severe dengue in children.

### 4.1. Cancer Biology

HA is pivotal in cancer biology, contributing to proliferation, metastasis, and multi-drug resistance (MDR). HA receptors, such as CD44, activate receptor tyrosine kinase pathways involving PI3K. This activation recruits guanine exchange factors (GEFs), which then indirectly trigger downstream GTPases like Rho and Ras within the Rho- and MAPK-signaling pathways [129].

#### 4.1.1. Phosphoinositide 3-Kinase (PI3K) Pathway

The PI3K pathway is essential in cancer cell proliferation, glucose metabolism, invasion, metastasis, angiogenesis, and survival [130]. PI3K consists of three classes, with class I responsible for regulating cell growth [131]. Class I is divided into two subtypes: class IA, activated by receptor tyrosine kinases, and class IB, activated by G-protein-coupled receptors. Akt, a serine-threonine kinase and key effector in the PI3K pathway, regulates the mTOR complex [132]. The PI3K pathway activation prevents cell death and enhances survival. Akt suppresses apoptosis by phosphorylating pro-apoptotic proteins like BAD and transcription factors FoxO1, FoxO3, and FoxO4 while promoting cell survival by phosphorylating cell survival proteins, including Bcl-2, Bcl-xL, MCL-1, A1, and BAG-1 [97]. The PI3K pathway has been recognized as the most commonly altered pro-cancer pathway in ovarian cancer [133], where the activation of PI3K, Akt, and mTOR accelerates cell proliferation and invasion [134]. Moreover, PI3K pathway activation is associated with therapy resistance in gastric, ovarian, uterine, and colon cancers.

HA molecules of 500 kDa and 1000 kDa trigger PI3K activation through CD44 in head, neck, and breast cancers, enhancing cell proliferation, motility, invasion, and therapy resistance [129]. In contrast, LMW-HA (~200 kDa), rather than HMW-HA (~1600 kDa), activates the PI3K pathway through RHAMM in JEG-3 choriocarcinoma cells, promoting cell migration [135]. These findings indicate that different cell types may react to different HA molecular weights through distinct receptors. HMW-HA (1500–1800 kDa) promotes cell growth in K562 leukemia cells (sensitive to vincristine) through the CD44-, MAPK-, and PI3K-signaling pathways. In Kv562 (resistant to vincristine) and lymphoma cells, HMW-HA activates only the PI3K pathway via RHAMM [94,136]. oHA prevents HA-CD44 interactions, thereby specifically reducing cell growth in K562 cells [136]. Furthermore, oHA inhibits PI3K pathway activation in lymphoma and colon carcinomas [94,109]. Despite studies investigating HA’s role in activating the PI3K pathway, there is conflicting evidence across different cell types. A more thorough analysis is required to elucidate the impact of HA molecular weight on the activation of the PI3K pathway in cancer cells.

#### 4.1.2. Rho GTPase Pathway

Rho GTPases are small GTP-binding proteins that regulate signal transduction and cell motility [137]. GEFs trigger conformational changes in Rho GTPases, exchanging GDP for GTP, which allows them to bind to target proteins [138]. The Rho GTPase family comprises 16 members, with Rho, Cdc42, and Rac being the most extensively studied [139]. Rho GTPase signaling enhances cancer cell motility by promoting actin polymerization, affecting cell stiffness, and contributing to MDR [139]. In ovarian cancer, Rho GTPase activation increases cell stiffness through actin polymerization, causing cisplatin resistance, which Rho inhibition can reverse. HA-induced activation of Rho GTPases has been observed in breast, head, neck, and ovarian cancers. In HSC-3 head and neck cancer cells, 500 kDa HA binds to CD44, triggering downstream RhoC and Rho-kinase activation, which promotes cell growth, migration, invasion, and cisplatin resistance [129]. In MDA-MB-231 breast and SKOV-3 ovarian cancer cells, HA molecules of 500 kDa and 1000 kDa indirectly activate Rho pathways through CD44, promoting cell growth and migration. In SKOV-3 ovarian cancer cells, 500 kDa HA activates Cdc42, while 1000 kDa HA activates Rac [140]. Rho GTPase activation by 500 kDa and 1000 kDa HA is confirmed, but the role of HA molecular weight in activating this pathway in cancer cells is yet to be studied.

#### 4.1.3. MAPK Pathway

The MAPK pathway, involving RAS GTPase, activates kinases RAF, MEK, and extracellular signal-regulated kinases 1 and 2 (ERK1/2), driving cell survival, proliferation, differentiation, and apoptosis [141]. Several MAPK proteins are associated with therapy resistance in cancer. The p38-MAPK pathway enhances trastuzumab resistance in breast cancer, whereas the MEK-ERK pathway contributes to saracatinib resistance in ovarian cancer [142,143]. A study on rabbit adipose-derived stem cells shows that HA induces ERK activation, with higher molecular weights (80–2000 kDa) promoting phospho-ERK activation [144]. Unlike ~1600 kDa HA, LMW-HA (weight unspecified) activates the MAPK pathway in JEG-3 choriocarcinoma cells via RHAMM, indicating a receptor or cell type-specific effect of HA [135]. In SKOV-3 cells, 1000 kDa of HA facilitate the CD44v3-Vav2 complex formation, activating Ras signaling [140]. Another study in CaOV-3 and SKOV-3 ovarian cancer cells shows that EGF triggers MAPK/ERK signaling to enhance cell migration, with HA and CD44 involvement in CaOV-3 cells [145]. In HSC-3 head and neck cancer cells, 500 kDa HA facilitates EGFR-CD44 complex formation, activating Ras GTPase, Raf, and ERK, which enhances cell invasion and migration [146]. Overall, evidence linking HA molecular weight to MAPK pathway activation in cancer is limited, with activation influenced by cell type and HA receptor interactions. Additionally, HA’s interaction with CD44 can facilitate complex formation with other receptors, potentially influencing the effects of different HA molecular weights.

## 5. The Nervous System

HA, as a key component of the brain’s ECM, significantly contributes to maintaining homeostasis in neuronal tissue by affecting cell differentiation, migration, proliferation, and other cellular behaviors. HA is predominantly synthesized by neurons and astrocytes, with astrocytes being responsible for producing HMW-HA [147]. Different sizes of HA display unique biological activities. HMW-HA inhibits the production of pro-inflammatory mediators such as cytokines and chemokines to reduce inflammation in different tissues. In contrast, LMW-HA promotes the production of these mediators, thereby inducing inflammatory processes [6,148,149]. HMW-HA also reduces the proliferation of various neural cell types [148,150,151,152,153], and both HMW-HA and HA fragments can regulate cell differentiation within the central nervous system (CNS) [152,153,154].

HA interacts with multiple receptors, affecting various biological processes, with CD44 and RHAMM, the most common receptors for HA. CD44 is a cell surface receptor present in glial and neuronal cells, and its interaction with HA affects cell behavior in neural tissue under both normal and pathological conditions [152,155,156]. For instance, HA-CD44 coupling in astrocytes triggers the Rac1-dependent PKN (protein kinase N-γ) pathway, resulting in cytoskeletal activation and increased migration of astrocytes under normal conditions [157]. The interaction between HA and CD44 regulates cell behaviors by activating Src family kinases (SFKs) and the focal adhesion kinase (FAK) cascade. Unlike CD44, RHAMM is found in the cytoplasm, nucleus, and on the cell surface [155]. When RHAMM interacts with HA, it can activate the SFK and FAK cascade and trigger Ras-signaling pathways to regulate cell behavior under normal physiological conditions, such as migration and proliferation [47,155].

The HA signaling can be altered by its association with other ECM components and covalent modifications. Some protein complexes, including the proteoglycans aggrecan, brevican, and neurocan, and link proteins like Bral1, Bral2, and HAPLN1, interact with HA in the nervous system ECM [158,159]. HA is also found within proteoglycans that accumulate around myelinated fibers and form specialized lattice-like ECM structures called perineuronal nets (PNNs). These nets envelop neuronal cells and play a vital role in brain development and neuroplasticity [160,161,162]. The covalent modifications to HA, such as the addition of IαI-HC catalyzed by TSG-6, can alter the signaling properties of HA. These changes influence neuroinflammation, HA crosslinking, and the interactions between HA and its receptors [163]. Furthermore, these extensive HA–protein complexes help to regulate nervous system functions, including cell adhesion, migration, and neurite outgrowth.

### 5.1. Neurodevelopment

HA is present in the CNS at the earliest stages of development, beginning when the neural plate folds into the neural tube. Although the precise amounts of HA in any area of the CNS or peripheral nervous system (PNS) have not yet been quantified at any life stage, HA levels peak in the embryonic nervous system and then decline into adulthood. Treating embryos with hyaluronidase to digest HMW-HA caused delays in neurulation, demonstrating HA’s role in some of the earliest morphogenic processes during neurodevelopment.

After neurulation, the ventricular zone, where neural stem cells (NSCs) undergo self-renewal, contains the lowest concentration of HA [153]. HA receptors, such as CD44 and RHAMM, are also expressed in the ventricular zone. RNA analyses of the developing CNS suggest that cell–ECM interactions via ECM receptors may be essential for the proliferative capacity of NSCs [155]. The complexity of this research is evident in ongoing investigations. The intermediate zone, where stem cells differentiate into neuroblasts, has higher HA concentrations [153]. However, it remains undetermined whether increased HA induces differentiation or creates a permissive environment that allows differentiating cells to begin migrating.

HA is essential for proper neocortical folding during the later stages of neurodevelopment [164]. Whether the matrix HA undergoes phase separation to guide molecular interactions is unknown. Indeed, HA plays a role in extending neurites and correctly guiding axons, such as those of retinal neurons in the optic pathway, as well as migratory cells in the developing cerebellum [165]. Additional research is necessary to fully comprehend how alterations in HA and HA-mediated signaling influence CNS morphogenesis.

In addition to its role in early CNS development, HA is also essential for early events in PNS development. After neurulation, HA is present surrounding the neural tube, notochord, and neural crest [159]. Neural crest cells express HAS1 and HAS2, with this expression being downregulated during migration [166]. These cells also express CD44 and RHAMM [155,167]. HAS expression, particularly HAS2, as well as HA, is present along the migratory paths of neural crest cells [159]. Knockdown of CD44 impairs neural crest migration, and silencing HAS2 during neural crest migration leads to migratory defects [167,168]. Taken together, HA can initiate neural crest cell migration for differentiation. Later, HA regulates neural progenitor cell proliferation and promotes neuronal differentiation, migration, and formation of cortical layers, including differentiation into neurons, Schwann cells, and other peripheral nerve cell populations.

### 5.2. Adult Neurogenic Niches

Adult neurogenic niches, such as the subgranular zone (SGZ) of the hippocampal dentate gyrus and the subventricular zone (SVZ), contain NSCs that undergo self-renewal and asymmetric division. These processes are crucial in generating new neurons and glial cells. HA, a key extracellular component in both of these niches [152,169], plays a significant role in this generation. HA signaling, mediated through receptors like CD44 and TLR2, is crucial in regulating stem cell proliferation and differentiation [152,170]. The loss of CD44 results in hippocampal dysfunction and deficits in spatial learning and memory [171]. This underscores the vital role of HA signaling in adult neurogenesis and in behaviors that rely on the integration of new neurons into neural circuits throughout life.

Moreover, HA signaling is linked to cell migration in the adult nervous system. The rostral migratory stream, a unique pathway that directs neuronal precursors from the subventricular zone (SVZ) to the olfactory bulb, exhibits elevated HA levels. The detection of RHAMM in migrating immature neurons within the rostral migratory stream indicates that interactions between HA and RHAMM are essential for these cells to migrate correctly [169]. These novel findings herald exciting opportunities for advancing research in neuroscience and cell biology.

In vitro research utilizing 3D HA hydrogels has demonstrated that the stiffness of the ECM can affect the characteristics of neural stem and progenitor cells. For instance, by adjusting HA concentration to create hydrogels that imitate the elasticities of both developing and adult brains, it was found that elasticity similar to the developing embryonic brain allows for neuronal progenitor migration and branching. In contrast, higher elasticities resembling the adult brain do not support these processes [172]. Moreover, 3D hydrogels with elevated HA concentrations have been found to reduce stem cell proliferation and differentiation [172]. The stiffness of the ECM can also enhance the proliferation, migration, and branching of one cell lineage over another [173]. It can even guide stem cell differentiation into various lineages [174]. Gradients in the ECM might direct cell migration from regions of lower to higher stiffness [175].

While how HA affects ECM stiffness in CNS tissues remains unclear, HABPs like TSG-6 may contribute to the modulation of HA crosslinking and subsequent changes in ECM stiffness. The size of the HA component can directly affect ECM stiffness, either on its own or by modulating TSG-induced crosslinking of HA strands. These findings indicate that variations in extracellular HA concentration during development can trigger cell responses by altering the mechanical properties of the ECM and interacting with HA receptors. Adjusting the mechanical properties of the ECM by varying HA size and concentration could also contribute to the regulation of neurogenesis in adult neurogenic niches.

### 5.3. Perineuronal Nets (PNNs)

HA is a widespread element of the ECM in the adult CNS, present in gray and white matter. HA provides a dynamic structural network to support and facilitate cell function and signaling. Perineuronal nets (PNNs) are specialized extracellular matrix structures that predominantly surround inhibitory neurons in the central nervous system (CNS) [173]. They have been identified as crucial regulators of synaptic plasticity and neuronal excitability.

It serves as a scaffold within the ECM, contributing to a more diffuse distribution essential for developing PNNs around neurons in gray matter [158,160,176,177]. HA also plays a critical structural role around myelinated axons and axon fiber tracts in white matter, as well as in the PNS [160,161,176]. It appears that phase separation contributes in part to HA concentration differences across various brain regions, as evidenced by studies revealing unique HA staining patterns in the somatosensory and piriform cortices, and the cerebellum [178]. HA works together with chondroitin sulfate proteoglycans (CSPGs) and link proteins for the formation of PNN [179]. Initially, it was believed that HA was synthesized by glia or by a combination of glia and developing neurons. However, other studies have shown that developing neurons alone express HAS enzymes and are responsible for constructing PNNs [177]. Glial cells and neurons may produce different components that collaborate in the formation of fully functional PNNs in various regions of the brain [162].

Cultured developing neurons gradually form PNNs over several days, replicating the slow development and maturation of these structures in the developing CNS [158,176,177,180,181]. In the developing CNS, the formation of PNNs occurs in a region-specific pattern that is associated with functional maturation [180]. PNNs are primarily located around parvalbumin-expressing inhibitory neurons, and it is believed that the synthesis and maturation of PNNs result in reduced synaptic plasticity [177,181]. This may occur by limiting the lateral diffusion of receptors, such as AMPA receptors, to specific regions on neuron cell bodies, including the postsynaptic densities [182]. Studies have shown that removing PNNs through HYAL treatment enhances excitatory neuronal activity and alters receptor subunit expression to patterns typically seen during critical periods [181]. Removing PNNs by HYAL treatment also elevates seizure-like activity of neurons in vitro [183] and raises seizure rates in animal models [184]. Interestingly, the prefrontal cortex of individuals with schizophrenia shows a reduction in PNNs [185]. These findings imply that developmental impairments in HA synthesis, resulting in reduced PNN formation, may contribute to CNS disorders like epilepsy and schizophrenia.

### 5.4. Homeostasis in Nervous Tissue

The ability of HA to impact the behavior of glial cells is an intriguing field of study. For example, astrocytes express HAS enzymes and produce HA [169]. The involvement of HA in regulating astrocyte migration and morphology occurs via interactions with CD44 and RHAMM. In vitro studies show that interactions between astrocytes and HA activate Rac-1 signaling, which promotes astrocyte migration in response to injury, such as a scratch wound in a cell monolayer culture [157]. In a mouse model, cortical ischemic lesions in the affected ipsilateral hemisphere have been shown to increase HA expression, potentially facilitating HA/RHAMM-mediated astrocyte migration in the peri-infarct zone [169]. Rodent oligodendrocyte progenitor cells (OPCs) treated with HA in vitro exhibit greater migration distance and speed than untreated controls. Moreover, this migratory response is blocked by inhibiting CD44 with a functional antibody. In vivo, the migration of OPCs transplanted into an inflamed spinal cord (induced by zymosan) is suppressed by pre-incubation with the CD44 blocking antibody [186]. HA’s potential to influence glial cell behavior opens exciting new possibilities for research and discovery.

Interactions between HA and cell receptors also play a role in regulating the proliferation of CNS cell types. In vitro studies on astrocytes indicate that HA is crucial in the formation of harmful glial scars by affecting astrocyte proliferation [148,150]. HMW-HA inhibits astrocyte proliferation in vitro, whereas its degradation into LMW-HA enhances astrocyte proliferation in a rat SCI model. Although the molecular weight of HA was not specified, rodents with cortical defects that received an intracortical injection of 3% HA gel showed a reduction in astrocyte proliferation compared to controls treated with saline solution alone [187]. During late embryonic and postnatal development, HA produced by astrocytes may contribute to ECM formation and reduce the proliferation of adjacent astrocytes, as well as neural stem cells in neurogenic niches. HA/CD44 interactions influence neural stem cell populations [152]. Neural stem cells expressing CD44 were isolated from the rodent subgranular zone and cultured using BrdU staining to mark proliferation. HA treatment significantly decreased BrdU incorporation in cultured neural stem cells compared to untreated controls, suggesting that HA regulates proliferation. LMW-HA decreases the proliferative capacity of mouse embryonic-derived neural progenitor cells by interacting with TLR-2 receptors in vitro [170].

Age-related alterations in HA content in the brain have been reported, while the conformational changes in HA remain largely unidentified [152]. In aging brains, the upregulation of HA synthesis has been observed in various cell types, including astrocytes and microvascular cells. The cause of this increased HA synthesis and its physiological consequences remain unclear. HA is associated with cell quiescence in NSCs and glial cells in vitro and in vivo [148,150,151,152,153]. Like NSCs, extracellular HMW-HA appears to serve as a quiescence signal for astrocytes both in vitro and in vivo, as well as for O2A progenitors that differentiate into type II astrocytes [148,150,151,172]. Considering that increased HA levels typically lead to cell quiescence, this suggests a decrease in cell proliferation. In neurogenic niches like the SGZ, the potential effect of elevated HA content on the significantly reduced neurogenesis observed in aging brains represents an important area for further investigation [188]. While the chemical gradients keep changing under the localized microenvironment in vivo, the exposure of the HA polyanionic charges is the key to regulating cell proliferation and behavioral alterations. Nonetheless, much less knowledge in this regard has not been established.

HA, a multifaceted regulator of receptor functions in both CNS and PNS neurons, presents an intriguing challenge for neuroscientists and researchers. The direct interactions between HA and the transient receptor potential cation channel subfamily V member 1 (TrpV1, the capsaicin receptor) expressed on dorsal root ganglion neurons lead to a reduction in receptor activation and sensitization, thereby diminishing responses to capsaicin and heat [189]. The depletion of HA by administering hyaluronidase to the footpad of rats increases pain sensitivity in a CD44-dependent manner, indicating that HA’s effects extend to the sensory nerve endings [190]. Additionally, HA has been shown to influence the activity of CNS neurons. Exogenous hyaluronidase decreases L-type voltage-gated calcium channel activity in hippocampal neurons and promotes AMPA receptor trafficking [182,191], suggesting that PNNs or similar structures may dynamically regulate neuronal activity. Importantly, the correct trafficking and function of glutamate transporters in astrocytes are also dependent on HA [192].

### 5.5. Injury and Nervous System Diseases

Whether the high-molecular-weight HA is protective against the progression of brain diseases or neurodegeneration remains controversial. The intact HA long cables and matrices may undergo alterations during brain trauma, dementia, ischemia, and inflammatory demyelinating diseases like multiple sclerosis (MS). In response to trauma, HMW-HA can modulate astrocyte proliferation, glial scar formation, and microglial activation [148,150,187]. Furthermore, HMW-HA affects the function of the vascular endothelial cell barrier and plays a role in lymphocyte extravasation and the initiation of experimental autoimmune encephalomyelitis (EAE), a mouse model of MS [193,194]. In pathological conditions, HA fragments have a wide range of effects, including the modulation of angiogenesis, neural stem cell proliferation, and the differentiation of neural progenitor cells [154,170]. Despite the plethora of HA effects in physiology and pathology, the nature of HA polymers in vivo is unknown, which requires further investigation. For example, can HA long cables undergo decoupling during inflammation, and does the short-chain HA disrupt the coupled state of HA long chains to balance the inflammatory settings?

Interplay between hyaluronan, chondroitin sulfates, and other proteoglycans is the key to the integrity of the ECM. The presence of HA and its binding proteins in the glial scars creates a barrier that limits the spread of inflammatory cells while also hindering axonal regeneration. Other binding proteins, like TSG-6, are upregulated in response to brain injury and Alzheimer’s disease. TSG-6 induces reactive astrocytes in a traumatic brain injury and inhibits microglial activation in vitro. TSG-6 also covalently modifies HA by forming HA-HC complexes, thereby altering HA interactions with receptors like CD44 and other signaling pathways during nervous system insults. TSG-6-catalyzed attachment of HC to HA molecules is reversible in HMW-HA, whereas HC binds irreversibly to smaller HA fragments (4–12 oligosaccharide units) [163]. IαI suppresses HA crosslinking by TSG-6 [195], implying that TSG-6 regulates HA degradation during injury and disease progression.

It has been assumed that HA and its binding proteins play a protective role in trauma by reducing neuronal damage from inflammation and promoting tissue healing and regeneration. For example, in white matter disorders like MS and vanishing white matter disease, the accumulation of HMW-HA and HA fragments in plaques has been shown to impede the maturation of OPCs and hinder functional remyelination [154,196]. In an EAE mouse model of MS, HA accumulates in demyelinated lesions [147]. The accumulation of HMW-HA and its production by CD44-positive astrocytes suppress remyelination in the spinal cord sections of the tested mice [147] and in patients with vanishing white matter disease [196]. Tissue studies from patients with vanishing white matter disease and other white matter disorders show that elevated levels of HA and CD44-positive cells result in defective maturation of glial cells through the activation of the TLR-2-signaling pathway [147,154,196]. HA stimulation also activates the TLR-4-signaling pathway, which plays a key role in the proliferation and differentiation of glioblastoma stem-like cells. During differentiation, glioblastoma stem-like cells increase TLR-4 expression, secrete HA, and activate the TLR-4-NFκB-signaling pathway. Activation of this pathway enables glioblastoma stem-like cells to sustain their proliferative state, suggesting that HA plays a role in the tumorigenic potential of these cells in vitro [197]. In support of this study, the proliferation of glioblastoma multiforme cultures is enhanced when exposed to higher concentrations of HA, as indicated by the increased size of cultured neurospheres [156].

## 6. Hyaluronan in Complement C1q-Regulated Inflammation and Alzheimer’s Disease

Native high-molecular-weight HA, even at high concentrations, is poor in blocking complement activation in vitro [198,199,200]. Worse still, native HA enhances cancer growth in vivo [5]. The accumulation of high-molecular-weight HA can occur in many diseased brains. As an initiator of the complement activation cascade, C1q is elevated in the brain and cerebrospinal fluid of Alzheimer’s disease (AD). Excessive C1q-mediated complement activation leads to HA degradation, neuronal damage, synapse loss, dysfunctional glial cells, and the recruitment of neutrophils to form neutrophil extracellular traps (NETs) [201,202,203]. The anti-C1q antibody has been postulated to be of therapeutic potential in treating AD and other neurodegenerative diseases. The binding of HA to the globular head of C1q (gC1q) affects cell proliferation, immune modulation, and cancer progression [201]. Also, HA binds the gC1q receptor (gC1qR) (also known as HABP1 or p32), thereby modulating cancer progression, metastasis, and immunotherapy [201]. How the HA-C1q axis is associated with the progression of AD is largely unknown.

### 6.1. The Controversial Role of C1q in Regulating Cell Survival

Whether C1q supports cell survival or death remains a matter of controversy. Human DU145 prostate cancer cells express functional WWOX (WWOXf) [204,205]. C1q signals the cell membrane WWOX to induce apoptosis, independently of complement activation of the classical pathway [206]. WWOX is a known tumor suppressor and is a risk factor for Alzheimer’s disease [5,29,204,205,206]. In contrast, C1q and HA support the growth, migration, and adherence of malignant pleural mesothelioma [201]. Intriguingly, HA-bound C1q is also capable of modulating HA synthesis. Malignant cancer cells do not express functional WWOX or are deficient in WWOX expression (WWOXd) [204,205,206]. We hypothesize that the C1q-pY33-WWOX axis suppresses cancer growth, including glioblastoma, neuroblastoma, and other cancer cells. pY33-WWOX is an activated form that maintains cellular physiological function and induces apoptosis when overexpressed [204,205,206]. In stark contrast, pS14-WWOX promotes the progression of Alzheimer’s disease and cancer cell growth [207]. Whether C1q and HA enhance the effects of pS14-WWOX remains to be established.

### 6.2. HA Reverses the C1q-pY33-WWOX Signaling: A Role in Alzheimer’s Disease?

When the WWOXf DU145 cells are cultured in complement C1q- or C6-depleted human serum (∆C1q or ∆C6), these cells exhibit constitutive activation of JNK1 (c-Jun N-terminal kinase 1) (Figure 4A) [206]. No effect was observed for ∆C9 human serum or under serum-free conditions. Normal human serum (NHS) sustains a basal level of JNK1 activation, suggesting that complements C1q and C6 can suppress JNK1 activation. Under similar conditions, ∆C1q or ∆C6 serum cannot activate STAT3 in DU145 cells. Notably, HA significantly induces STAT3 activation when cells are cultured in the ∆C1q serum, compared to serum-free, NHS, ∆C6, or ∆C9 serum (Figure 4B). That is, in the absence of C1q, HA can support the activation of the prosurvival STAT3. STAT3 is activated by upstream FGFR and JAK kinase. Once activated, STAT3 dimerizes and relocates to the nucleus to mediate gene expression, thereby facilitating cell proliferation and survival [208]. Aberrant activation of STAT3 is associated with the development and progression of various cancers.

Without C9, p53 becomes activated and relocates to the nuclei of DU145 cells (Figure 4C). HA reverses p53 activation. Similar results were observed under serum-free conditions. P53 does not undergo nuclear relocation when cultured in ∆C6 serum, whereas HA induces p53 activation (Figure 4C). When DU145 cells are cultured in ∆C1q, ∆C6, ∆C7, ∆C8, or ∆C9 serum, or serum-free conditions, no activation or nuclear localization of ERK and WWOX is observed (Figure 4D).

Overall, complement proteins provide a regulatory network of signal transduction pathways. Serums C1q and C6 restrict JNK1 activation in DU145 cells. HA activates STAT3 in C1q-free serum, suggesting that C1q limits STAT3 activation. C9 limits p53 activation, and HA reverses the inhibitory effect. Notably, C6 is needed for p53 activation, and HA induces p53 activation in ∆C6 serum. C1q activates WWOX and ERK to induce apoptosis [206]. Additional work is required to establish the complement-regulated signaling network, both in the presence and absence of HA, in normal neurons, liver cells, lung cells, and other primary cell types versus cancer cells. Accordingly, the generation of amyloid beta in the assay system allows for the assessment of HA, C1q, and other complement proteins in the regulation of AD progression.

### 6.3. HA-Activated-Signaling Pathways: Crosstalk with C1q-WWOX Pathway

High-molecular-weight native HA induces the activation of Smad2, 3, and 4, ERK, JNK, pS46-p53, Hyal-2, WWOX, pY287-WWOX, and WWOX2 isoform in a panel of WWOXf cells, including HCT116, DU145, MCF7, Jurkat, MEF Wwox wild type, and L929S cells [5]. Similarly, HA causes the activation of Smad2, 3, and 4, as well as ERK, in WWOX-deficient cells, such as 4T1, MDA-MB-231, MDA-MB-435S, and MEF Wwox knockout cells [5]. The crosstalk in signaling between C1q and HA involves ERK and WWOX, suggesting that both proteins, when they become aberrant, are needed for the progression of Alzheimer’s disease.

### 6.4. C1q Induces the Formation of WWOX-Containing Microvilli Clusters for Apoptosis

Total internal reflection fluorescence (TIRF) microscopy reveals the dynamic changes of the protein expression on the cell membrane and cytoskeletal areas [206]. When DU145 cells are transiently overexpressed with EGFP-WWOX, these cells touch and adhere to the glass surface only by EGFP-WWOX-energized clustered microvilli (Figure 4F; see green puncta). The cells’ ventral areas are detached from the surface. C1q rapidly increases the formation of clustered microvilli between cells. In EGFP-expressing cells, no microvilli are observed, and C1q does not induce microvilli formation. During a prolonged exposure to C1q for 24 h, the cells undergo apoptosis (Figure 4F).

### 6.5. C1q and Degraded HA Cause Neurodegeneration Under Inflammatory Conditions

Under inflammatory conditions, C1q upregulates the expression of hyaluronidases HYALs to degrade native HA. The degraded HA is likely to have coupled long chains because these degraded hyaluronidases have poor activity in anti-inflammation. C1q and the degraded HA are proposed to bind WWOX on the surface of microvilli of the adhered cells (Figure 4G). As a result, degradation of the membrane APP (amyloid precursor protein) occurs, thereby leading to the generation of amyloid beta aggregates. Whether membrane-anchored WWOX binds the aggregated amyloid beta on the extracellular matrix remains to be established. HYAL-2 anchors the membrane-bound WWOX [5].

### 6.6. Disordered HA Is Potent in Blocking Cancer Growth and Probably Retarding Neurodegeneration

Native HA becomes a potent inducer of apoptosis when the signaling complex HYAL-2/WWOX/SMAD4 is overexpressed [5] (Figure 4H). Whether the HYAL-2/WWOX/SMAD4 complex is overexpressed in vivo is unknown. Hyaluronidase-degraded HA is poor in anti-inflammation. This is due to the interchain coupling in the degraded HA, which has concealed polyanionic charges (Figure 4H) [198,199,200]. Sonicated HA has a disordered conformation, exposing polyanionic charges that affect many biomolecules. The disordered HA strongly blocks complement activation, compared to native HA [198,199,200]. Similarly, sonicated HA has an exposure of polyanionic charges that block the function of many proteins [5]. Furthermore, sonicated HA activates HYAL-2+ CD3- CD19- Z cells to kill cancer cells [5]. Activated Z cells are able to kill cancer cells [5].

## 7. Discussion

High-molecular-weight HA has been widely utilized to treat injured knee joints. HA is also used to repair retinal damage. HA is an essential component in cosmetics. HA is a crucial regulator for maintaining normal physiology and immune response. HA of 10 million Daltons is potent in blocking cancer growth in naked mole rats [8]. However, HA of 2–4 million Daltons or less promotes cancer growth in mammals. The smaller HA chains generated during inflammation are believed to enhance cancer or disease-mediated inflammation. It is generally agreed that the larger the HA size, the more effective HA is in anti-inflammation. The smaller HA supports inflammation. Whether the phase separation of proteins in the HA matrix participates in the complement-mediated inflammation and progression of neurodegenerative diseases remains to be established.

What has been overlooked is that the anti-inflammatory effect of HA is due to the exposure of its polyanionic charges. Without exposure, HA-mediated anti-inflammatory effects against complement activation are poor [198,199,200]. Similarly, the anticancer activity of a highly condensed HA matrix is also poor. The postulation that high-molecular-weight HA is anti-inflammatory and low-molecular-weight HA is pro-inflammatory is not correct.

Focus should be placed on manipulating the exposure of HA polyanionic charges, so that the anti-inflammatory function of HA can be greatly boosted. Digestion of native HA with hyaluronidase cannot result in disordered conformation with exposure of polyanionic chains. Similarly, exposure of native HA to UV leads to degradation but does not show a fully disordered conformation. Sonication is a reliable approach to generating disordered HA for treating cancer and probably neurodegeneration.

There are many approaches attempting to utilize HA for design as therapeutic agents or as drug carriers. However, most fall short by failing to recognize the fundamental concept of HA being readily subjected to conformational changes in its changing microenvironment. HA chains, either long or short, are subjected to conformational changes upon exposure to the chemical gradients in the microenvironment. Consequently, functional HA becomes dysfunctional when changes occur in the surrounding chemistry of the microenvironment. For example, when HA is suspended in phosphate-buffered saline and sonicated at 53 kHz for 8 h, it becomes highly potent in blocking cancer growth [5]. In stark contrast, if HA is suspended in Milli-Q water for sonication, the resulting HA fails to block cancer growth. Clearly, subtle changes in the chemicals surrounding HA long chains affect not only the conformation of HA but also its functional activity. Thus, when considering drug design using HA, the fundamental concept of HA conformational changes must be considered.

Current strategies for modulating HA-related pathways include enzymatic degradation, receptor inhibition, targeted drug delivery, and synthetic HA mimetics. Hyaluronidase-based therapies, such as pegylated recombinant human hyaluronidase (PEGPH20) [209], have been explored to degrade tumor-associated HA, improving drug delivery and enhancing chemotherapy efficacy in HA-rich tumors. Another approach involves targeting HA receptors, such as CD44 and RHAMM, with monoclonal antibodies and small-molecule inhibitors to suppress cancer cell migration and invasion. Additionally, HA-based nanocarriers have been developed for drug delivery, utilizing receptor-mediated endocytosis to improve drug accumulation in tumor cells and inflamed neural tissues while minimizing off-target effects. Synthetic HA fragments and competitive inhibitors have also been investigated for their potential to disrupt oncogenic and neuroinflammatory pathways by modulating ECM remodeling, immune responses, and tumor plasticity.

Additionally, targeting HA metabolism in a personalized manner could enhance therapeutic precision, as identifying patient subgroups based on HA metabolism and receptor expression profiles may lead to more effective treatments with fewer side effects. Another critical challenge is overcoming resistance mechanisms in HA-targeted therapies, as some cancers develop compensatory pathways to evade HA inhibition. Exploring HA’s role in neural regeneration and neuroprotection is also an emerging area of interest, as HA has been implicated in neurogenesis, synaptic plasticity, and brain repair. Furthermore, the integration of HA in biomaterial scaffolds for regenerative medicine presents exciting opportunities for developing advanced HA-based hydrogels, nanoparticles, and scaffolds to improve wound healing, neural repair, and tissue regeneration. Addressing these challenges and opportunities will be essential for optimizing HA-targeted therapies in oncology and neurology.

Finally, outstanding questions to be addressed in advancing our knowledge of HA are listed in Table 1. It is essential to define the mobility, transportation, and aggregation of small molecules such as glucose and ions, as well as proteins, in normal and inflamed HA matrices. From these, we should be able to monitor the phase-separation event in the HA matrix. In this case, we can limit complement activation, as well as the activation-related inflammation and neurodegeneration. Further, we can design HA matrices that support cell stemness, proliferation, and differentiation without causing accidental cell death. Finally, we must examine the possibility of using disordered HA in controlling inflammation in lesions of cancers and neurodegenerative diseases.

## 8. Conclusions

HA plays a crucial role in maintaining normal physiology; alterations in the HA matrix may lead to disease progression. We consider the potential effect of phase separation in the HA matrices that affects the mobility and behavior of small molecules, proteins, and cells. The occurrence of protein aggregation is likely. For example, degraded HA binds C1q to activate the subsequent classical complement pathway, which may involve the aggregation of the protein components in the classical pathway due to phase separation. Under inflammatory conditions, the degraded HA may signal with the overexpressed HYAL-2/WWOX/SMAD4 complex, a complex known for its role in cell signaling and apoptosis, which induces apoptosis. TGF-β competes with the HYAL-2/WWOX/SMAD4-signaling pathway, although its effect is unknown. Of particular importance is that highly disordered HA conformation confers cancer suppression and probable retardation of neurodegeneration. The induction of a disordered conformation of HA should be well established.

## 9. Patents

United States Patent US9375447B2: Modified hyaluronan and uses thereof in cancer treatment, by Nan-Shan Chang and Wan-Pei Su, National Cheng Kung University, Taiwan.

## Figures and Tables

**Figure 1 ijms-26-05132-f001:**
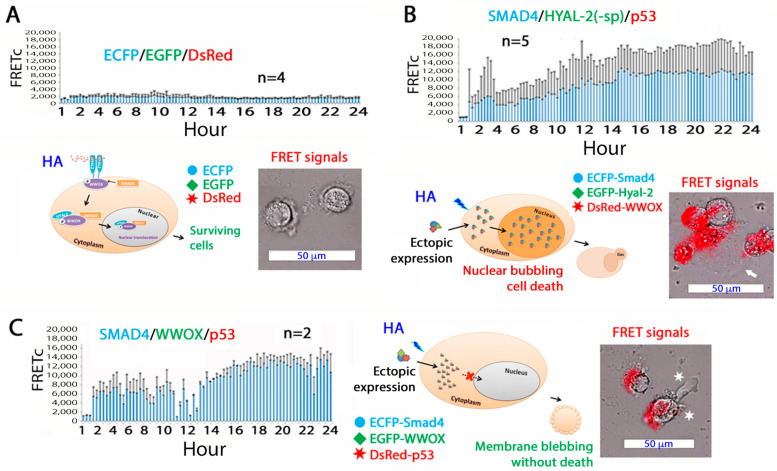
Native HA induces cell death via signaling with overexpressed WWOX, HYAL-2, and SMAD4. (**A**,**B**) WWOX functional prostate DU145 cells were transfected with three indicated expression plasmids tagged with ECFP, EGFP, or DsRedl and then added native HA. Time-lapse FRET microscopy revealed that HA increased the binding strength of SMAD4/HYAL-2(-sp)/WWOX (red fluorescence for FRETc signals) and nuclear-dependent bubbling cell death (**B**). No complex formation occurred with the empty vectors alone (**A**). (**C**) HA induced the complex formation of SMAD4/p53/WWOX. The complex caused membrane blebbing without rendering apoptosis. X-axis for time in hour; FRET = Förster resonance energy transfer or fluorescence resonance energy transfer. The FRET images were corrected for background fluorescence from an area free of cells. Y-axis for FRETc = the spectrally corrected FRET concentration [5].

**Figure 2 ijms-26-05132-f002:**
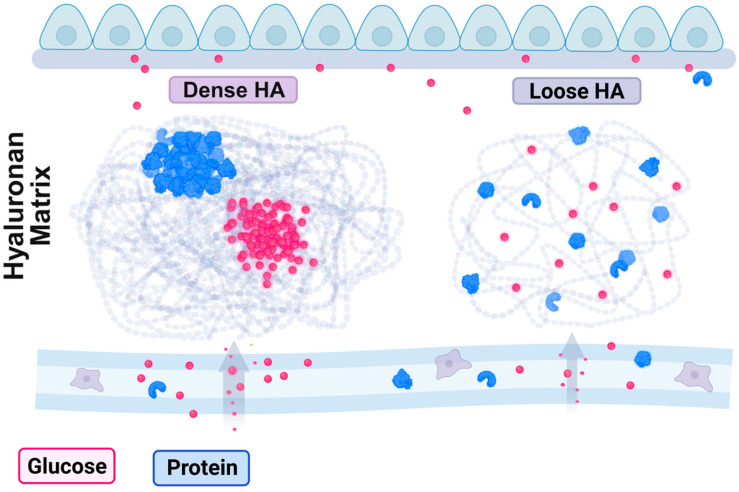
HA matrix induces phase separation and controls molecular movement. Two matrices with high and low densities of HA are shown. Glucose is released from the bloodstream and moves into the HA matrices. Phase separation may occur for glucose and a protein in the high-density HA matrix. The phase-separated large-sized materials are expected to have a retarded movement and slow uptake by surrounding cells. Glucose may have enhanced or retarded intake by cells depending upon the size of the separated materials.

**Figure 3 ijms-26-05132-f003:**
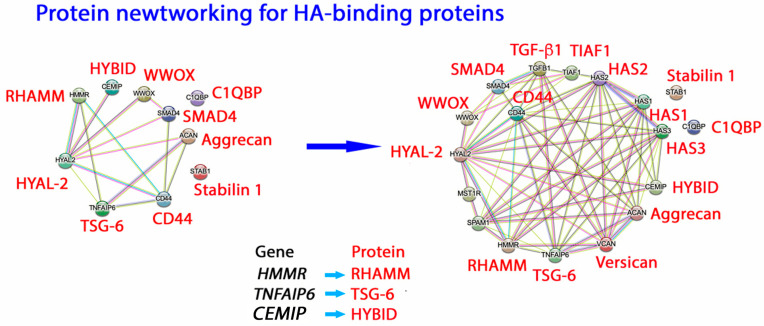
HA-binding protein interacting network. HYAL-2 is regarded as the center of the interacting network. The initial layer reveals that HYAL-2 may have functional and/or physical interactions with target proteins, including RHMM, HYBID, WWOX, C1QBP, SMAD4, Aggrecan, Stabilin 1, CD44, and TSG-6. In the second layer, the presence of TGF-β1, TIAF1, HAS1, HAS2, HAS3, and Versican is added. The functional property of each protein is discussed in the text shown below.

**Figure 4 ijms-26-05132-f004:**
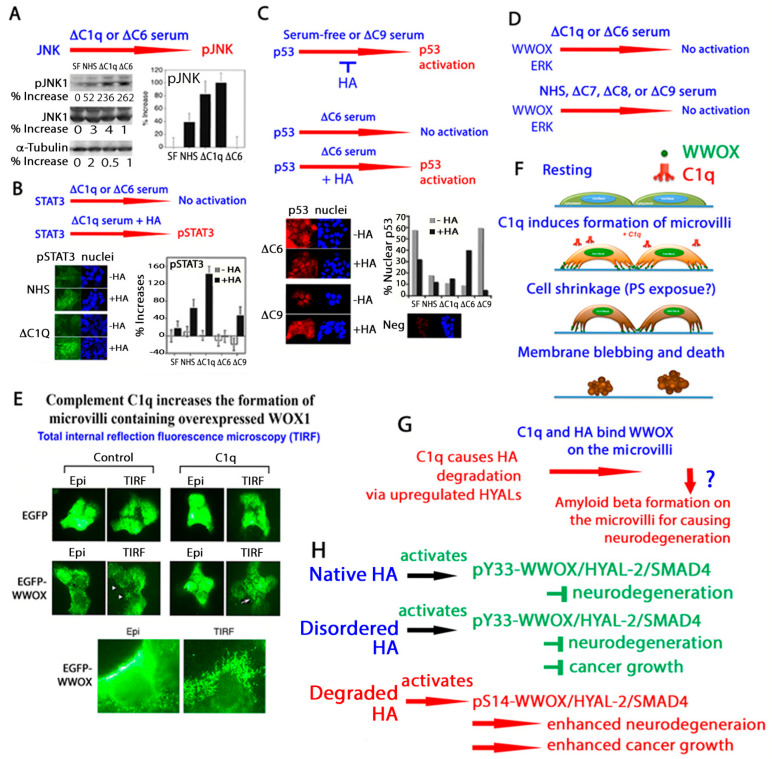
C1q induces clustered WWOX-microvilli formation on the cell surface: potential role of HA and C1q in neurodegeneration. (**A**) Human ∆C1q or ∆C6 serum induces JNK activation in DU145 cells. (**B**) ∆C1q or ∆C6 serum cannot activate STAT3, whereas HA significantly induces STAT3 activation in the ∆C1q serum. (**C**) Without C9 or NHS, p53 is activated in DU145 cells, and HA reverses this effect. In contrast, HA induces p53 activation in ∆C6 serum. (**D**) No activation of ERK and WWOX is observed in ∆C1q, ∆C6, ∆C7, ∆C8, or ∆C9 serum, or under serum-free conditions. (**E**,**F**) EGFP-expressing DU145 cells adhere flatly to the cover glass surface during overnight culture. Epi: epifluorescence. Transiently overexpressed EGFP-WWOX induces clustered punctate formation on the ventral cell surface (see arrowheads; 600× magnification). C1q further increases the punctate formation and causes apoptosis. (**G**) C1q upregulates HYALs to degrade native HA. The degraded HA exhibits poor anti-inflammatory activity. C1q and the degraded HA probably bind WWOX on the surface of microvilli. (**H**). Hyaluronidase-degraded HA is chain-coupled and exhibits poor anti-inflammatory properties. Sonicated HA has a disordered conformation to exert cancer suppression via activated HYAL-2+ CD3- CD19- Z cells to kill cancer cells [5].

**Table 1 ijms-26-05132-t001:** Outstanding questions for HA in drug development.

**Molecular Nature and Behavior**
How to delineate the molecular mobility, transportation, and aggregation in normal and inflamed HA matrices.
Elucidation of the puzzles would facilitate our understanding of biological reactions. For example, the interaction of C1q with HA in the matrix in synovial fluid or brain tissue can be a suitable model for study.
**Phase separation in monitoring molecular behavior and interactions**
How to control the effect of phase separation in the HA matrix to limit complement activation, and the activation-related inflammation and neurodegeneration.
Understanding the HA matrix’s role in causing phase separation for both small and large molecules would enable us to prepare an appropriate HA matrix and control the molecular interactions within it. For example, glucose may be highly concentrated in a particular area of an HA matrix. Their transportation and uptake by surrounding synovial or brain cells would explain the phase-separated molecules in supporting normal physiology or disease development.
**Molecular design**
How to design HA matrices suitable for supporting cell stemness, proliferation, and differentiation without accidental cell death.
To maintain the stability of an HA matrix without further metabolic degradation, mixing of the physically modified long-chain and short-chain HA samples via heat treatment or sonication is expected to stabilize the HA matrix.
**Disordered HA suppression of the HA/C1q/pS14-WWOX for neurodegeneration**
How to utilize sonication to generate disordered HA to block neurodegeneration caused by degraded HA interacting with C1q and pS14-WWOX.
The pathway for the formation of the disease-promoting HA/C1q/pS14-WWOX complex is unknown. Using disordered HA is likely to suppress WWOX phosphorylation of S14, thereby preventing the progression of Alzheimer’s disease.

## Data Availability

Available from Nan-Shan Chang laboratory.

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
