# Peer review of "Hyaluronan: An Architect and Integrator for Cancer and Neural Diseases"

_ijms, 2025, doi:10.3390/ijms26115132_

Round 1
Reviewer 1 Report
Comments and Suggestions for Authors
Comments for authors
The manuscript under revision entiteled "The role of hyaluronan in cancer progression and neural diseases" presents a comprehensive review of the biological roles of hyaluronan in cancer progression and neural diseases, detailing its metabolism, signaling pathways and interactions with binding proteins. The scientific relevance of the topic is high, as hyaluronan is increasingly recognized as a key regulator in various pathological processes, particularly in oncology and neurobiology.
However, despite the thorough literature review and detailed mechanistic insights, the manuscript has several issues that need to be addressed before it can be considered suitable for publication.
Major Comments
1- The manuscripts seem to lack critical analysis. Include a section discussing knowledge deficits, controversies and future perspectives in the field. Also improve the structure to avoid excessive repetition and improving readability.
2- While the title suggests an equal focus on cancer and neural disorders, the majority of the manuscript focuses on cancer. Increse the discussion of the role of hyaluronan in neurodegenerative diseases, brain injury and neuroinflammation to balance the review. Otherwise, the title should be changed and limited to the topic of the paper.
3- The manuscript lacks schematic diagrams or illustrations that would enhance understanding of complex molecular pathways. Figures summarizing hyaluronan metabolism, receptor signaling and disease mechanisms should be included for clarity and to improve the quality of the review.
4- Some sections have a limited number of bibliographic references and the references are not up to date. In general, there are only 3 reference from 2022, one from 2023 and no references from 2024 are included. An update of the bibliography is needed for a revision to be published in 2025.
5- The conclusion should be thoroughly revised. The section does not concisely restate the main findings of the study. The discussion focuses strictly on the ultrasound experiments without broader implications for therapeutic applications. It is also not recommended to include references in the conclusions. Please review thoroughly.
Although the manuscript is relevant, it needs significant improvement in terms of novelty, structure, balance and data presentation. Incorporation of the above suggestions will increase its impact and relevance for a high quality journal such as IJMS. I suggest that it be considered for acceptance in IJMS after major revisions.
Comments on the Quality of English LanguageSee comments above.
Author Response
Comments for authors
The manuscript under revision entiteled "The role of hyaluronan in cancer progression and neural diseases" presents a comprehensive review of the biological roles of hyaluronan in cancer progression and neural diseases, detailing its metabolism, signaling pathways and interactions with binding proteins. The scientific relevance of the topic is high, as hyaluronan is increasingly recognized as a key regulator in various pathological processes, particularly in oncology and neurobiology.
However, despite the thorough literature review and detailed mechanistic insights, the manuscript has several issues that need to be addressed before it can be considered suitable for publication.
Answer: Thank you. We agree.
Major Comments
1- The manuscripts seem to lack critical analysis. Include a section discussing knowledge deficits, controversies and future perspectives in the field. Also, improve the structure to avoid excessive repetition and improve readability.
Answer: Thank you. As you requested, we have added brief introductory paragraphs to each section, describing what we will write about, current knowledge in the field, areas of deficiency, and areas of controversy.
Key changes in this manuscript are: 1) Native high-molecular-weight HA cannot induce apoptosis. However, if cells have an overexpressed HYAL-2/WWOX/SMAD4 complex, the native HA induces apoptosis. We addressed the potential role of native HA and the apoptosis signaling in an inflamed microenvironment; 2) HA-binding proteins are now well-connected via signaling networks. We addressed the competitive binding of HA and TGF-b to the membrane HYAL-2 and the downstream HYAL-2/WWOX/SMAD4 signaling; 3) The potential role of proteins or small molecules under phase separation in the HA matrices. We discuss the potential correlations with disease progression; 4) We also describe the role of HA and complement C1q in Alzheimer’s disease. The connection of HA/C1q with WWOX is also thoroughly discussed. WWOX is a risk factor for Alzheimer’s disease; 5) HA conformational changes and relationship with cancer suppression.
Additionally, we have discussed the critical role of high-molecular-weight HA in tissue protection, blocking inflammation, cancer growth, and neurodegeneration. These scenarios are controversial and may not be correct. As requested, we have now polished essentially every section to remove redundancy. Finally, at the beginning of this manuscript, we have included the research history of hyaluronan. We also mentioned the recent passing of Dr. Gerard Armand, one of the pioneers in HA research, as well as his significant contributions to the HA field.
2- While the title suggests an equal focus on cancer and neural disorders, the majority of the manuscript focuses on cancer. Increase the discussion of the role of hyaluronan in neurodegenerative diseases, brain injury, and neuroinflammation to balance the review. Otherwise, the title should be changed and limited to the topic of the paper.
Answer: Thank you. We decided to change the original title. The new title is “Hyaluronan: an Architect and Integrator for Cancer and Neural Diseases.” We have emphasized the link between cancer and neurodegeneration, increased the scope of neurodegeneration, and discussed conformationally disordered hyaluronan in treating cancer and neurodegeneration, such as Alzheimer’s disease.
3- The manuscript lacks schematic diagrams or illustrations that would enhance understanding of complex molecular pathways. Figures summarizing hyaluronan metabolism, receptor signaling and disease mechanisms should be included for clarity and to improve the quality of the review.
Answer: As requested, we have added four figures: 1) HA-regulated cell death via Hyal-2/WWOX/SMAD4 signaling, 2) HA matrix induces phase separation and controls molecular movement, and the implications for controlling homeostasis and disease progression, 3) HA binding protein networking and their implications in homeostasis and disease progression, 4) HA signaling with complement C1q, C6, and C9 in cells, and HA/complement C1q signaling for WWOX-regulated Alzheimer’s disease progression.
4- Some sections have a limited number of bibliographic references and the references are not up to date. In general, there are only 3 reference from 2022, one from 2023 and no references from 2024 are included. An update of the bibliography is needed for a revision to be published in 2025.
Answer: As requested, we have updated references wherever needed. Some old references are crucial. We need to respect the original findings, also.
5- The conclusion should be thoroughly revised. The section does not concisely restate the main findings of the study. The discussion focuses strictly on the ultrasound experiments without broader implications for therapeutic applications. It is also not recommended to include references in the conclusions. Please review thoroughly.
Answer: As requested, we have rewritten and shortened the Conclusion without references. The Discussion has also been revised.
Reviewer 2 Report
Comments and Suggestions for Authors
This submission is a review of the role of hyaluronin (HA) in cancer and neural diseases. It describes the basic biology of HA, its synthesis and catabolism. This is followed by interactions of HA with other molecules in its role as a receptor ligand and in cell adherence. Roles in cancer include proliferation, metastasis, and signal transduction. The roles of HA in neural development and neural homeostasis are discussed. Finally, the discussion revolves around the potential for therapeutic application of HA-targeted therapies in cancer and neural disease.
The content is well-documented; the text is well written and organized.
One suggestion for clarity -
Line 982 –“ While there are many approaches out there to utilize HA for design as therapeutic agents or as drug carriers, many approaches fall short of including the fundamental concept of HA readily subjected to conformational changes in its changing microenvironment.” Confusing. Would read better as…
“There are many approaches attempting to utilize HA for design as therapeutic agents or as drug carriers, however most fall short by failing to recognize the fundamental concept of HA being readily subjected to conformational changes in its changing microenvironment.”
Author Response
This submission is a review of the role of hyaluronin (HA) in cancer and neural diseases. It describes the basic biology of HA, its synthesis and catabolism. This is followed by interactions of HA with other molecules in its role as a receptor ligand and in cell adherence. Roles in cancer include proliferation, metastasis, and signal transduction. The roles of HA in neural development and neural homeostasis are discussed. Finally, the discussion revolves around the potential for therapeutic application of HA-targeted therapies in cancer and neural disease.
The content is well-documented; the text is well written and organized.
One suggestion for clarity -
Line 982 –“ While there are many approaches out there to utilize HA for design as therapeutic agents or as drug carriers, many approaches fall short of including the fundamental concept of HA readily subjected to conformational changes in its changing microenvironment.” Confusing. Would read better as…
“There are many approaches attempting to utilize HA for design as therapeutic agents or as drug carriers, however most fall short by failing to recognize the fundamental concept of HA being readily subjected to conformational changes in its changing microenvironment.”
Answer: Thank you. The sentence has been replaced (Lines 1199 to 1202). Please also note that we have made the dramatic changes in the manuscript, including the abstract, main text, discussion, conclusion, four new figures, and Table I for Outstanding Question.
Round 2
Reviewer 1 Report
Comments and Suggestions for Authors
In this latest version, the authors have addressed the reviewers' comments and have made significant improvements to the manuscript. The authors have redefined the concept of the review by incorporating new authors and modifying the title and content.
The revised review manuscript provides a comprehensive, detailed, and conceptually rich overview of hyaluronan (HA) and its roles in cancer and neurological diseases. The authors have made a commendable effort to integrate diverse molecular mechanisms, signaling pathways, and clinical implications of HA biology into a coherent account. Notably, the analysis of phase separation, HA-interacting proteins, and HA conformational dynamics provides novel perspectives that were seldom addressed in previous reviews.
The manuscript incorporates updated literature and addresses emerging concepts, such as HA-induced apoptotic signaling, competitive interactions between receptors (e.g., with HYAL-2 and TGF-β), and the relevance of HA assemblies in disease progression. The tribute to Professor Armand is appropriate and respectfully incorporated into the scientific narrative.
For all these reasons, I recommend its publication in IJMS.
Reviewer 2 Report
Comments and Suggestions for Authors
This submission is a review of the role of hyaluronin (HA) in cancer and neural diseases. This version contains a significant amount of new material and figure have been added for clarity. These are positive changes.
The content is well-documented; the text is well written and organized.
Previous comments have been addressed.